# Transformer-Based Spatial-Temporal Counterfactual Outcomes Estimation

He Li [* 1]   Haoang Chi [* 2 1]   Mingyu Liu [1]   Wanrong Huang [1]   Liyang Xu [1]   Wenjing Yang [1]

## Abstract

The real world naturally has dimensions of time and space. Therefore, estimating the counterfactual outcomes with spatial-temporal attributes is a crucial problem. However, previous methods are based on classical statistical models, which still have limitations in performance and generalization. This paper proposes a novel framework for estimating counterfactual outcomes with spatial-temporal attributes using the Transformer, exhibiting stronger estimation ability. Under mild assumptions, the proposed estimator within this framework is consistent and asymptotically normal. To validate the effectiveness of our approach, we conduct simulation experiments and real data experiments. Simulation experiments show that our estimator has a stronger estimation capability than baseline methods. Real data experiments provide a valuable conclusion to the causal effect of conflicts on forest loss in Colombia. The source code is available at this URL.

## 1. Introduction

Causal inference plays a vital role in various fields, such as epidemiology (Lawlor et al., 2008; Robins et al., 2000) and economics (Baum-Snow & Ferreira, 2015; Dague & Lahey, 2019). Understanding and utilizing causality helps us reveal the mechanisms of the physical world, predict the occurrence of events in the real world, and manipulate their outcomes. Counterfactual outcome prediction (Prosperi et al., 2020; Wang et al., 2025) is a promising direction in causal inference. In practice, some causal problems have spatial-temporal attributes, such as the well-known "Butterfly Effect": a butterfly flapping its wings in Brazil may eventually lead to a tornado in the United States. As we know, time and space are closely entangled and inseparable (Einstein, 1915; 1922), and the physical world naturally possesses spatial-temporal attributes. Thus, it is crucial to perform counterfactual outcomes estimation with spatial-temporal attributes.

However, the classical causal inference frameworks (Pearl, 2010; Rubin, 1974) cannot be directly applied to estimate counterfactual outcomes with spatial-temporal attributes. A few studies Christiansen et al. (2022); Papadogeorgou et al. (2022) have initially explored causal inference methods for spatial-temporal data based on classical statistical models. Christiansen et al. (2022) propose to use the structural causal models for spatial-temporal data. However, in their setting, current outcomes are mainly affected by current treatment. Thus, their estimands primarily focus on the causal effects simultaneously, which does not fully address the issue of temporal carryover effects. Papadogeorgou et al. (2022) extend the potential outcomes framework to the spatial-temporal setting. Although they propose the spatial-temporal potential outcomes framework, they don't explicitly propose a method for the computation of propensity scores with spatial-temporal attributes, which is essential for outcomes estimation. Besides, they use classical statistical methods, such as kernel methods, which could suffer from complicated data patterns. The kernel methods rely on correctly specified kernel functions and smoothing parameters to ensure optimal performance. Thus, previous works still have limitations in performance and generalization.

In this work, we propose a novel framework for the estimation of counterfactual outcomes with spatial-temporal attributes using deep learning. An overview of the studied problem is shown in Figure 1, the studied spatial-temporal data is the time series of point patterns. Our objective is to estimate the expected number of occurrences of the outcome events within a specific region under a counterfactual treatment assignment strategy. The counterfactual treatment assignment strategy refers to a strategy that does not exist in the observational data. In other words, we aim to investigate "what will happen to outcome events when we employ other treatment strategies?". Accordingly, we propose deep-learning-based estimators within a spatial-temporal causal inference framework motivated by Papadogeorgou et al. (2022). The proposed estimators account for the effects of past treatments on current outcomes, thus capturing the

---

[*]Equal contribution  [1]College of Computer Science and Technology, National University of Defense Technology, Changsha, China  [2]Intelligent Game and Decision Lab, Academy of Military Science, Beijing, China. Correspondence to: Wenjing Yang <wenjing.yang@nudt.edu.cn>.

*Proceedings of the 42$^{nd}$ International Conference on Machine Learning*, Vancouver, Canada. PMLR 267, 2025. Copyright 2025 by the author(s).

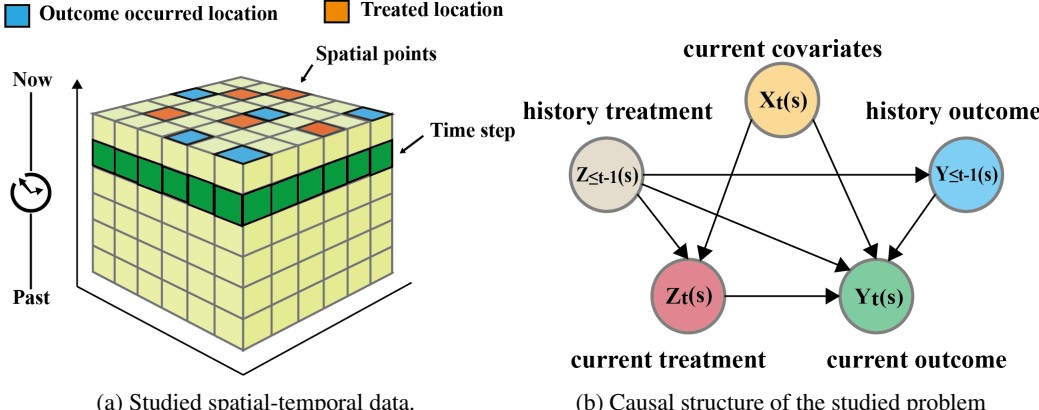

(a) Studied spatial-temporal data.

(b) Causal structure of the studied problem

*Figure 1.* Overview of the studied problem. The left figure demonstrates the studied spatial-temporal data. Each layer in the cube represents a spatial point pattern in a time step. The blue blocks represent the outcomes that occurred, while the orange blocks indicate the treated locations. The point patterns at multiple time steps from the past to the present constitute spatial-temporal data, which can be viewed as high-dimensional tensors. The right figure illustrates the causal structure of the studied problem, where $s$ refers to spatial location and $t$ represents time.

temporal causal effects. Besides, under mild assumptions, our estimators are based on inverse probability weighting and two innovations. First, based on convolutional neural networks (CNNs), we propose an efficient method to compute propensity scores when both treatment and covariates are high-dimensional, such as point pattern series. Second, we employ a Transformer-based model to estimate the intensity functions of point processes that characterize spatial-temporal data. Moreover, our deep-learning-based estimators within this framework exhibit excellent statistical properties, such as consistency and asymptotic normality.

To validate the effectiveness of our approach, we conduct simulations and real data experiments. Simulation experiments demonstrate that our deep-learning-based estimator exhibits lower estimation bias than the four baseline methods. Real data experiments study the causal effect of conflicts on forest loss in Colombia from the year 2002 to the year 2022. The results conclude that the longer duration and more intensity of conflicts will cause more forest loss in response.

Overall, our contributions are summarized as follows:

- We study counterfactual outcomes estimation with the spatial-temporal attribute, a more general setting, and propose an effective deep-learning-based solution.

- We propose an efficient CNN-based method to address the calculation of propensity scores when both treatment and covariates are high-dimensional, such as point patterns. In addition, we use the Transformer to model the intensity functions of point processes, used to characterize spatial-temporal data.

- We empirically demonstrate the effectiveness of our

approach by both simulated and real experiments. The real data experiments study the cause-and-effect of conflicts on forest loss in Colombia, revealing a valuable conclusion: longer and more intense conflicts lead to an increase in forest loss.

## 2. Related Works

### 2.1. Temporal Causal Inference

Temporal causal inference aims to investigate the causal relationships among time series data. Several studies have focused on estimating counterfactual outcomes or treatment effects of time-varying treatments. Examples include Granger Causality (Granger, 1969), Marginal Structural Models (MSMs) (Robins et al., 2000), and Recurrent Marginal Structural Networks (RMSNs) (Lim, 2018). However, time-varying confounders may lead to bias in the estimation. To address the problem of the time-varying confounders, several deep learning-based methods, such as Causal Transformer, were proposed (Bica et al., 2020a;b; Li et al., 2020; Liu et al., 2020; Melnychuk et al., 2022; Vo et al., 2021). There exist methods targeting other types of sequential data, such as natural language (Chi et al., 2024) and temporal omics data (Zhang et al., 2024). While these methods provide strong theoretical foundations for temporal causal inference, their models do not include spatial dimensions.

### 2.2. Temporal Counterfactual Prediction

Temporal counterfactual prediction refers to performing counterfactual outcome prediction under time-varying settings. Seedat et al. (2022) propose TE-CDE, a neural-controlled differential equation approach for counterfactual outcome estimation in irregularly sampled time-series

data. Wu et al. (2024) introduce a conditional generative framework for counterfactual outcome estimation under time-varying treatments, addressing challenges in high-dimensional outcomes and distribution mismatch. El Bouchattaoui et al. (2024) propose an RNN-based approach for counterfactual regression over time, focusing on long-term treatment effect estimation.

### 2.3. Spatial Causal Inference

Spatial data is commonly found in the environment, such as air quality data in a region and the incidence rate of a disease in a region (Cressie & Wikle, 2015). Spatial data often includes spatial correlation and heterogeneity, which challenge treatment effect estimation (Akbari et al., 2023; Ning et al., 2018). To address the problem of spatial correlation, Jarner et al. (2002) eliminate the effect of known covariates by matching the treatment group and the control group and estimating the latent spatial confounding under this design. Concerning the heterogeneity treatment, Causal Forest (Wager & Athey, 2018) is a non-parametric random forest-based method for heterogeneity treatment effects estimation. The above methods ignore the temporal dimension, while we focus on spatial-temporal data.

### 2.4. Spatial-Temporal Causal Inference

Spatial-temporal data refers to the data consisting of spatial and temporal information (Cressie & Wikle, 2015). In the real world, spatial-temporal data, such as temperature variation over time in a region, is ubiquitous, and many of them are time series of remote-sensing images. However, there has been limited research on the causal inference of spatial-temporal data until now. Christiansen et al. (2022) extended the structure causal model to make it applicable to spatial-temporal data. However, in their settings, current outcomes are mainly affected by the current treatments, which does not fully solve the temporal carryover effect. Papadogeorgou et al. (2022) extended the potential outcome framework on spatial-temporal data and employed kernel functions to estimate the treatment effects. However, they didn't explicitly propose a method to compute propensity scores with spatial-temporal attributes, which is essential for treatment effects estimation. In practice, spatial-temporal causality is effective for better understanding multi-modal data, such as video (Li et al., 2023; 2022) and market data (Li et al., 2024).

## 3. Background

Now we introduce the important concepts used in this framework and estimators.

**Spatial Point Patterns.** The point patterns describe the locations where events occur within a given region and are

often characterized using point process models (Baddeley et al., 2008). The rectangle layer with colored blocks in Figure 1a illustrates an example of a spatial point pattern. We employ spatial point patterns to describe the treatments and outcomes in a time step.

**Spatial Point Process and Intensity Function.** The spatial point process is a statistical model that describes the spatial distribution of the locations of events in a given region. Formally, let $\mathcal{N}(\cdot)$ denote the counting measure, and $D_s \subset \mathbb{R}^d$ is a measurable subset of $\mathbb{R}^d$. A spatial point process can be defined as a collection of random variables $\{\mathcal{N}(A)|A \subset D_s\}$, where $\mathcal{N}(A)$ represents the number of events occurring in a specific spatial region $A$. An important characteristic of the spatial point process is the expected number of events occurring within a region $A$, i.e., $E[\mathcal{N}(A)]$. It can be calculated by an intensity function, which represents the average density of events occurring within a unit region. Let $s$ denote a location and $ds$ denote a small region located at location $s$ with area $v(ds)$, the definition of the intensity function $\lambda(\cdot)$ is:

$$\lambda(s) = \lim_{v(ds) \to 0} \frac{E[\mathcal{N}(ds)]}{v(ds)}.$$

Then we have $E[\mathcal{N}(A)] = \int_A \lambda(s)ds$, $A \subset D_s$. Intensity functions are crucial to characterize spatial point processes, which can be utilized to generate spatial point patterns.

**Spatial Poisson Point Process.** An important spatial point process is the spatial Poisson point process, which is also the primary focus of this paper. We continue to use $\lambda(s)$ to denote the intensity function of a Poisson point process. An important property of a Poisson point process is $\mathcal{N}(A) \sim$ Poisson$(\lambda_A)$, and $\lambda_A = \int_A \lambda(s)ds$. Poisson$(\cdot)$ denotes the Poisson distribution. According to the properties of the Poisson distribution, $E[\mathcal{N}(A)] = \lambda_A$. The detailed proof is in Appendix E.

## 4. Method

We first introduce the potential outcome framework for spatial-temporal data. Based on this framework, we describe the estimands of interest. Finally, we derive the estimators and implement them with deep learning models.

### 4.1. Potential Outcome Framework for Spatial-Temporal Data

Before formalizing our problem setting and estimands, we introduce the potential outcome framework for spatial-temporal data (Papadogeorgou et al., 2022). Specifically, denote $Z_t(s)$ as a binary treatment variable, and $Z_t(s) = 1$, $Z_t(s) = 0$ represent the presence and absence of treatment at time $t$ and location $s$, respectively. Let $\gamma = \{1, 2, \ldots, T\}$ denote the time index set, and $\Omega$ denote a spatial region.

Then $Z_t = \{Z_t(s)|s \in \Omega\}$ is the collection of binary treatment variables of all the locations in the region $\Omega$. $Z_t$ can be regarded as a spatial point pattern generated by a spatial point process, with $Z_t(s) = 1$ indicating the presence of a point and $Z_t(s) = 0$ indicating its absence. $Z = \{Z_t|t \in \gamma\}$ is the collection of treatments in time index set $\gamma$. By taking a subset of $Z$, we construct a historical treatment set up to time $t$, i.e., $Z_{\leq t} = \{Z_1, Z_2, \ldots, Z_t\}$. By assigning $Z_{\leq t}$ as $z_{\leq t}$, we have the historical treatment realization $z_{\leq t} = \{z_1, z_2, \ldots, z_t\}$, where $z_1, z_2, \ldots, z_t$ are the assignments of $Z_1, Z_2, \ldots, Z_t$. Then, we denote $S_{z_t} = \{s \in \Omega|z_t(s) = 1\}$ as the set of locations that have received treatment in time $t$.

Similarly, we can formalize the outcome. Specifically, denote $Y_t(Z_{\leq t})(s)$ as a binary outcome variable, and $Y_t(Z_{\leq t})(s) = 1$, $Y_t(Z_{\leq t})(s) = 0$ represent the occurrence and non-occurrence of the target event at time $t$ and position $s$ under a treatment $Z_{\leq t}$, respectively. Let $Y_t(Z_{\leq t}) = \{Y_t(Z_{\leq t})(s)|s \in \Omega\}$ denote the potential outcome in a region $\Omega$ given the historical treatment $Z_{\leq t}$, by assigning $Z_{\leq t}$ as $z_{\leq t}$, and we can distinguish $Y_t(Z_{\leq t})$ from the notion of observed outcome $Y_t^{ob}(z_{\leq t}) = Y_t(Z_{\leq t})|_{Z_{\leq t}=z_{\leq t}}$. $Y_t^{ob}(z_{\leq t})$ represents a spatial point pattern generated by a spatial point process, with $Y_t(z_{\leq t})(s) = 1$ indicating the presence of a point and $Y_t(z_{\leq t})(s) = 0$ indicating its absence. Likewise, $Y_{\leq t}$ and $Y_{\leq t}^{ob}$ represent the potential outcome and observed outcome up to time $t$, respectively. $S_{Y_t^{ob}(z_{\leq t})} = \{s \in \Omega|Y_t^{ob}(z_{\leq t})(s) = 1\}$ is the collection of the locations whose target outcomes occur in time $t$.

Let $X_t$ denote the covariates at time $t$ and $X_{\leq t} = \{X_1, X_2, \ldots, X_t\}$ denote the covariates up to time $t$, and $x_{\leq t}\{x_1, x_2, \ldots, x_t\}$ is its realization. Finally, by combining the treatment, outcome, and covariates, we obtain the historical information. Let $h_{\leq t} = \{z_{\leq t}, Y_{\leq t}^{ob}, x_{\leq t}\}$ be the observed historical information up to time $t$, and $H_{\leq t} = \{Z_{\leq t}, Y_{\leq t}, X_{\leq t}\}$ be the potential historical information up to time $t$. Clearly, $h_{\leq t} \subset H_{\leq t}$. We summarize all notations in Appendix A.

## 4.2. Counterfactual Outcomes

We aim to estimate the expected number of outcome-occurring locations in an area under a counterfactual treatment assignment mechanism. First, we introduce the method for treatment intervention.

**Treatment Intervention.** We consider stochastic treatment intervention, making treatment follow a specific intervention distribution. Let $F_h(\cdot)$ denote a spatial point pattern distribution drawn from a spatial Poisson point process with intensity function $h$. Then $F_h(z_t)$ represents we assign treatment realization $z_t$ following the distribution $F_h(\cdot)$. For a historical treatment realization $z_{\leq t} = \{z_1, z_2, \ldots, z_t\}$, we can apply $F_h(\cdot)$ to the last element of $z_{\leq t}$ (the latest treatment), i.e., $z_{\leq t}(F_h) :=$

$\{z_1, z_2, \ldots, F_h(z_t)\}$, to intervene the treatment. Similarly, let $F_H(\cdot) = F_{h_1}(\cdot) \times F_{h_2}(\cdot) \times \ldots \times F_{h_M}(\cdot)$ denote a joint distribution with $M$ independent ones, and $z_{[t - M + 1, t]} = (z_{t-M+1}, z_{t-M+2}, \ldots, z_t)$ denote the last $M$ elements of $z_{\leq t}$. We can apply $F_H(\cdot)$ to $z_{[t - M + 1, t]}$ to obtain $z_{\leq t}(F_H) := \{z_1, \ldots, z_{t-M}, F_{h_1}(z_{t-M+1}), \ldots, F_{h_M}(z_t)\}$. It is noted that **the proposed treatment intervention method is an improvement of the method in** (Papadogeorgou et al., 2022). Specifically, Papadogeorgou et al. (2022) assume the intervention distributions within $F_H$ are all the same ($F_h$), while we allow the distributions in $F_H$ to be diverse and change with time (i.e., $F_{h_t}$). Recall that $F_H$ denotes the intervened distributions of previous treatments. **Therefore, the estimands in our setting are more general than those in** (Papadogeorgou et al., 2022).

**Estimands.** Based on the treatment intervention method, we define the expected number of outcomes in time $t$ and a region $\omega$ as the estimand of interest $N_t^\omega(F_H)$. Under the intervention distribution $F_H(\cdot)$, we have

$$N_t^\omega(F_H) = \int_{Z^M} |S_{Y_t^{ob}(z_{\leq t}(F_H))} \cap \omega| dF_H(z_{[t - M + 1, t]}). \tag{1}$$

Furthermore, taking the mean values of $N_t^\omega(F_H)$ over time $t = M, M + 1, \ldots, T$, we obtain,

$$N_\omega(F_H) = \frac{1}{T - M + 1} \sum_{t=M}^T N_t^\omega(F_H). \tag{2}$$

It is noted that **the treatments of $M$ periods follow the intervention distribution $F_H$ we specified, which does not necessarily exist in the observable data. Thus, the proposed estimands are counterfactual.** To better demonstrate the estimands, we provide a running example that walks through each term of the estimands in Appendix B.

## 4.3. Assumptions

To identify the above estimands, we introduce the necessary mild assumptions.

**Assumption 1. (Unconfoundedness)** We assume that condition on $h_{\leq t}$, $Z_t$ is not dependent on $H_{\leq t}$: $Z_t \perp\!\!\!\perp H_{\leq t}|h_{\leq t}$. This assumption is also known as ignorability (Rubin, 1978), which is commonly used in causal inference, indicating the absence of unmeasured covariates.

**Assumption 2. (Poisson Assumption)** We assume that $Z_t$, $Y_t(Z_{\leq t})$ and $Z_t|h_{\leq t-1}$ are generated from spatial Poisson point processes. This assumption is reasonable since the spatial Poisson point processes are widely used to characterize the distribution of discrete events in a region (Cressie & Wikle, 2015).

Additional assumptions and their explanations can be found in Appendix C. We provide the proof of identifiability for the estimands in Appendix D.

## 4.4. Estimators

With the above assumptions, we derive estimators for the corresponding estimands. The proposed estimators are based on inverse probability weighting (IPW). As a key component of IPW, we first introduce the propensity score within the framework used in this work.

**Propensity Score.** The propensity score within this framework is the probability of treatment $Z_t$ given historical information $h_{\leq t-1}$, denoted as $e_t(z_t) = \mathbb{P}(Z_t = z_t|h_{\leq t-1})$.

**Counterfactual Probability.** We define the counterfactual probability of treatment $Z_t$ as $p_h(z_t)$, i.e., $p_h(z_t) = f_h(Z_t = z_t)$. $f_h(\cdot)$ denotes the probability density function of the intervention distribution $F_h(\cdot)$. A detailed explanation of why $p_h(z_t)$ is "counterfactual" is in Appendix H.

**Intensity Function.** In section 4.1, we have defined $Y_t^{ob}(z_{\leq t}) = \{Y_t^{ob}(z_{\leq t})(s)|s \in \Omega\}$. $Y_t^{ob}(z_{\leq t})$ is a discrete spatial point pattern generated by a spatial Poisson point process. We denote the intensity function of this latent spatial Poisson point process as $\lambda_{Y_t^{ob}(z_{\leq t})}(s)$, this intensity function is assumed to be continuous, serving as a spatial smoothing of $Y_t^{ob}(z_{\leq t})$.

**Inverse Probability Weighting Estimator.** With the above components, the inverse probability weighting estimator $\hat{Y}_t(F_H, s)$ is derived as follows:

$$\hat{Y}_t(F_H, s) = \prod_{j=t-M+1}^{t} \frac{p_{h_j}(z_j)}{e_j(z_j)} \lambda_{Y_t^{ob}(z_{\leq t})}(s). \quad (3)$$

$\hat{Y}_t(F_H, s)$ can be viewed as the intensity function of a spatial Poisson point process that generates $Y_t^{ob}(z_{\leq t}(F_H))$. According to the definition of intensity function in Section 3, the expected number of outcome-occur points in a region $\omega$ is obtained by integrating $\hat{Y}_t(F_H, s)$ over region $\omega$. Therefore, the estimator of $N_t^\omega(F_H)$ is shown as follows:

$$\hat{N}_t^\omega(F_H) = \int_\omega \hat{Y}_t(F_H, s)ds. \quad (4)$$

Then we obtain the estimator of $N_\omega(F_H)$, $\hat{N}_\omega(F_H) = \frac{1}{T-M+1} \sum_{t=M}^{T} \hat{N}_t^\omega(F_H)$.

We briefly outline the derivation of our estimator. It is built upon the Inverse Probability Weighting (IPW) approach from Marginal Structural Models (MSMs) (Robins et al., 2000). In particular, the term $\lambda_{Y_t^{ob}(z_{\leq t})}(s)$ in Eq. (3) denotes the intensity function of the observed outcome at time $t$ and location $s$, while the product $\prod_{j=t-M+1}^{t} \frac{p_{h_j}(z_j)}{e_j(z_j)}$ serves as a weighting factor analogous to that used in MSMs. The method extends the IPW-based weighting mechanism from MSMs to a spatial-temporal framework.

## 4.5. Theoretical Properties

Now we introduce the important theoretical properties of the propensity score and estimator.

**Proposition 1.** The propensity score is a balancing score. For any $t \in \gamma$,

$$Z_t \perp\!\!\!\perp h_{\leq t}|e_t(z_t).$$

**Proposition 2.** Dimensional reduction property of the propensity score: if $Z_t \perp\!\!\!\perp H_{\leq t}|h_{\leq t}$ then $Z_t \perp\!\!\!\perp H_{\leq t}|e_t(z_t)$.

**Proposition 3. (The consistency and asymptotic normality of the estimator)** Let $Var$ denote the Variance, $N_\omega(Y_t)$ denote the number of the observed outcome $Y_t$ in region $\omega$. If all assumptions hold, and $T \to \infty$, we have that

$$\sqrt{T}(\hat{N}_\omega(F_H) - N_\omega(F_H)) \xrightarrow{d} N(0, v),$$

where $v = lim_{T\to\infty} \frac{1}{T-M+1} \sum_{t=M}^{T} v_t$ and $v_t = Var[\prod_{j=t-M+1}^{t} \frac{p_{h_j}(z_j)}{e_j(z_j)} N_\omega(Y_t)|H_{\leq t-M}]$.

The proofs of the above propositions are in Appendix C.

## 4.6. Deep-Learning-Based Realization

This subsection introduces how we employ neural networks to realize the above estimators. Figure 2 shows the full architecture of our model.

### 4.6.1. CALCULATE THE PROPENSITY SCORE

**Challenges.** We first discuss the challenges that high-dimensional data brings and why the classical classification-based method cannot address this problem. In the classical setting, treatments take limited values (e.g., binary treatments). Thus, we can utilize covariates to train a classifier of treatments and employ this classifier to predict the propensity scores. However, in our setting, the treatment $Z_t$ is high-dimensional. Specifically, $Z_t = \{Z_t(s)|s \in \Omega\}$ contains all the binary treatment variables of locations in region $\Omega$. If there are 100 locations $s$, the $Z_t$ can take $2^{100}$ values. Then we need to train a classifier with $2^{100}$ classes, which is unacceptable. To address the challenges, we propose the following method.

**Dimension Reduction.** Since it's hard to compute the propensity scores of high-dimensional treatments directly, we utilize a dimension reduction map to project treatments into a low-dimensional space. Let $R(\cdot)$ denote the dimension reduction map. We define it as follows:

$$R(Z_t) = |\{Z_t(s); Z_t(s) = 1, s \in \Omega\}|. \quad (5)$$

$R(Z_t)$ maps $Z_t$ to a low-dimensional space, representing the count of treated locations in time $t$. We focus on the count of treated points when constructing estimators. Therefore, this dimension reduction map preserves the informa-

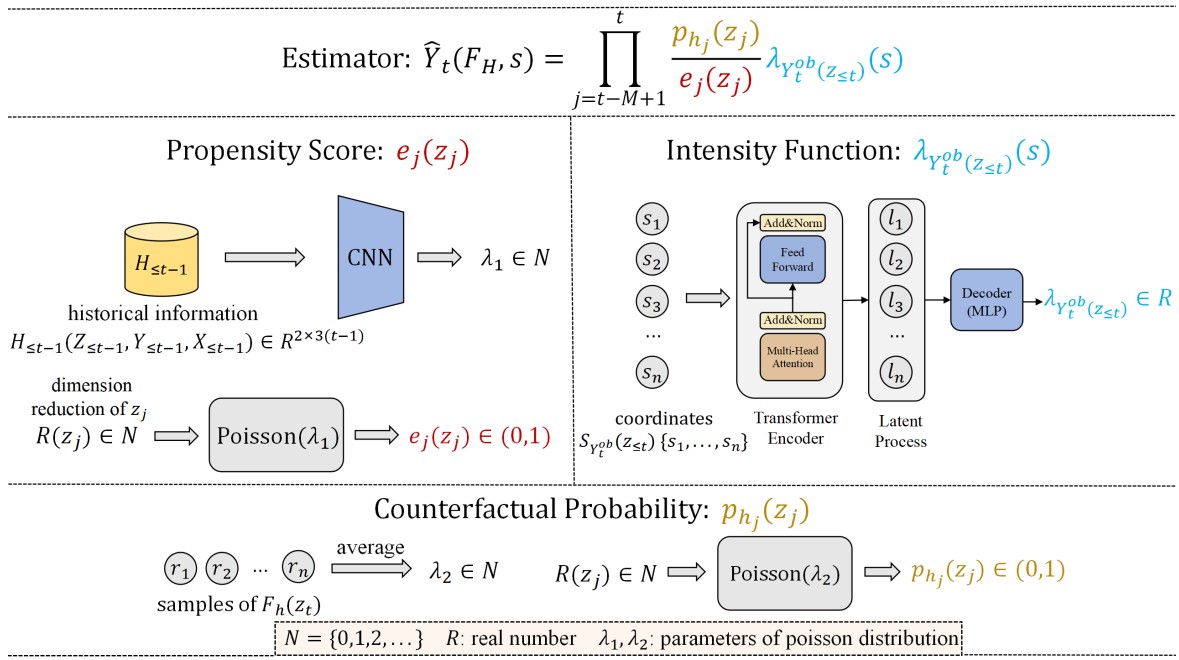

Figure 2. Full model architecture.

tion in the treatments while facilitating efficient computation of propensity scores.

After the dimension reduction of treatment, we express the propensity score in the following form:

$$e_t(R(z_t)) = \mathbb{P}(R(Z_t) = R(z_t)|h_{\leq t-1}). \tag{6}$$

Then we specify the distribution $\mathbb{P}(R(Z_t) = R(z_t)|h_{\leq t-1})$ to calculate the propensity score. According to Assumption 2, $Z_t|h_{\leq t-1}$ is generated by a spatial Poisson point process. With the properties of spatial Poisson point process in section 3, we have $R(Z_t)|h_{\leq t-1} \sim \text{Poisson}(\lambda_1)$. Thus, the task of calculating the propensity score is transformed into estimating the parameter $\lambda_1$ of the Poisson distribution $\text{Poisson}(\lambda_1)$.

**CNN-based Regression.** We formulate the estimation of $\lambda_1$ into a regression problem. Specifically, we employ the high-dimensional series $h_{\leq t-1}$ to regress scalar $R(Z_t)$. According to the definition of regression, such a regression model outputs $E(R(Z_t)|h_{\leq t-1})$[1]. **Since CNNs efficiently extract local features from high-dimensional tensors. They capture spatial hierarchies and patterns, making them ideal for data like 3D structures.** We employ CNNs as the regression model and use the MSE loss during training. Specifically, denote the output of CNNs as $output(\cdot)$, then $\text{MSE} = E(output(h_{\leq t-1}) - R(z_t))^2$. A detailed structure of the used CNNs is in Appendix K.

---

[1]Regression models can be seen as modeling the probability distribution of the dependent variable, with the predicted output corresponding to conditional expectation.

Consequently, according to the properties of the Poisson distribution, $E(R(Z_t)|h_{\leq t-1})$ is synonymous with the parameter $\lambda_1$ that governs $R(Z_t)|h_{\leq t-1}$. Detailed derivation can be found in Appendix E. Utilizing the Poisson distribution, the propensity score can be calculated by the following equation:

$$e_t(R(z_t)) = \frac{\lambda_1^{R(z_t)}}{R(z_t)!}e^{-\lambda_1}. \tag{7}$$

#### 4.6.2. CALCULATE COUNTERFACTUAL PROBABILITY

Analyzing the probability distribution of high-dimensional variables directly is challenging due to the "curse of dimensionality", which leads to data sparsity, increased complexity of relationships, and higher computational costs (Verleysen & François, 2005). Thus, we utilize the dimension reduction map to project $Z_t$ into a low-dimensional space. Consequently, the transformed counterfactual probability takes the form:

$$p_h(R(z_t)) = f_h(R(Z_t) = R(z_t)). \tag{8}$$

With the properties of spatial Poisson point processes in section 3, $R(Z_t)$ follows a Poisson distribution given by: $R(Z_t) \sim \text{Poisson}(\lambda_2)$. Then, the objective is transformed into estimating the parameter $\lambda_2$.

As shown in Figure 2, to achieve this goal, we draw samples from the Poisson point process $F_h(z_t)$ and estimate $E(R(Z_t))$ using the sample mean of $R(Z_t)$. With the property of Poisson distribution, $E(R(Z_t))$ is equivalent to $\lambda_2$,

the detailed derivation is in Appendix E. Leveraging the definition of Poisson distribution, the counterfactual probability is computed by the following equation:

$$p_h(R(z_t)) = \frac{\lambda_2^{R(z_t)}}{R(z_t)!} e^{-\lambda_2}. \tag{9}$$

### 4.6.3. TRANSFORMER-BASED INTENSITY FUNCTION ESTIMATION

As for the estimation of the intensity function, we draw inspiration from the maximum likelihood estimation (MLE). The core idea is to use the neural network to model $\lambda_{Y_t^{ob}(z_{\le t})}(s)$, and train this network to maximize the likelihood function. We define $net : \Omega \to R$ as the output of this neural network. The training objective is to minimize the following function:

$$- \sum_{i=1}^{|S_{Y_t^{ob}(z_{\le t})}|} \ln(net(s_i)) + \int_\Omega net(s)ds - \mathrm{KL}(q\|p), \tag{10}$$

where $s_i \in S_{Y_t^{ob}(z_{\le t})}$ is the coordinate in $S_{Y_t^{ob}(z_{\le t})}$, $q$ is the data distribution obtained from the Transformer encoder and Gaussian sampling, and $p$ is a prior standard Gaussian distribution. $\mathrm{KL}(q\|p)$ is the Kullback–Leibler divergence between two distributions $q$ and $p$. Note that $\ln(net(s_i))$ and $\int_\Omega net(s)$ are the components of the likelihood function of the spatial Poisson point process. A detailed derivation of the training objective is in Appendix F. An overall architecture of the network is depicted in Figure 2.

**Why choose the Transformer?** During training, we input coordinates sequence in $S_{Y_t^{ob}(z_{\le t})}$ into the network. Since the Transformer can capture long-term and high-order dependencies and meanwhile enjoy computational efficiency (Zuo et al., 2020), the ability to capture such dependencies creates more powerful models than RNNs, which facilitates the estimation of intensity function (Zhou et al., 2022; Zuo et al., 2020). Thus, we employ the Transformer to realize this network. In Section 5.5, through experiments, we further demonstrate the superiority of the Transformer model.

## 5. Experiments and Results

We evaluate our method on both synthetic and real datasets.

### 5.1. Datasets

#### 5.1.1. SYNTHETIC DATA

We employ the intensity functions of spatial Poisson point processes to generate synthetic spatial-temporal data. Generation details are in Appendix M.1.

#### 5.1.2. REAL WORLD DATA

**Forest Change Data.** The Global Forest Change Data is an annually updated global dataset capturing forest loss derived

from time-series images acquired by the Landsat satellite (Hansen et al., 2013). We consider forest loss events in Colombia from 2002 to 2022 and treat them as outcomes.

**UCDP Georeferenced Event Dataset.** The UCDP Georeferenced Event Dataset is a comprehensive dataset documenting global conflicts, providing details on parameters such as the temporal aspects and geographical coordinates of conflict events (Croicu & Sundberg, 2015). We exclusively examine conflict events transpiring within the territorial boundaries of Colombia and consider them as treatments. The chronological span of our analysis encompasses the years from 2002 to 2022.

A brief overview and detailed description of the real-world data are in Appendix L.

### 5.2. Baselines, Metrics, and Implementations

**Baselines.** The chosen baselines are state-of-the-art literature on the counterfactual outcomes estimation (Lim, 2018; Robins et al., 2000), and the heterogeneous treatment effects estimation (Wager & Athey, 2018). These are: MSMs (Robins et al., 2000), RMSNs (Lim, 2018), Causal Forest (Wager & Athey, 2018). We also employ a linear regression-based method (LR) as an additional baseline for comparison. Since these baselines cannot handle the high-dimensional series directly, we transformed all our synthetic data into scalar series to adapt the baselines to our setting. Appendix N details the baselines and adaptation.

**Metrics.** For synthetic experiments, we compute the true counterfactual outcomes as ground truth and employ the relative error rate (RER) between the estimation and the ground truth as a metric. Details of the ground truth are in Appendix M.2. The formula of RER is shown below:

$$\mathrm{RER} = \frac{|\text{Estimated Value - True Value}|}{|\text{True Value}|},$$

where $|\cdot|$ denotes the absolute value.

For real data experiments, since the true counterfactual outcomes of real data are unknown, we consider the consistency of our conclusions with existing literature on the effects of human conflicts on natural resources.

**Error Bars.** For each experimental setup, we conduct 20 independent runs. Reported values are the means, and error bars indicate the standard deviation ($\pm\sigma$) over the 20 runs.

**Implementations.** For CNNs in section 4.6.1, the details are epoch = 200, learning rate = 0.001, batch size = 64. For the Transformer-based neural network in Figure 2, the details are as follows: The number of blocks in the Transformer encoder is 8, the number of attention heads per layer is 16, the number of layers in the Multi-Layer Perceptron decoder (MLP) is 8, epoch = 300, learning rate = 0.0001.

*Table 1.* Real data experiments results. The data in the table are the estimated numbers of forest loss events.

|         | $c = 3$        | $c = 4$        | $c = 5$        | $c = 6$        | $c = 7$        |
|---------|----------------|----------------|----------------|----------------|----------------|
| $M = 1$ | $20.6 \pm 2.3$ | $20.5 \pm 2.2$ | $20.7 \pm 2.8$ | $20.5 \pm 1.9$ | $20.8 \pm 2.0$ |
| $M = 3$ | $21.5 \pm 1.4$ | $21.6 \pm 2.4$ | $22.3 \pm 1.9$ | $23.0 \pm 1.3$ | $23.3 \pm 1.9$ |
| $M = 5$ | $22.4 \pm 2.3$ | $22.9 \pm 1.8$ | $23.6 \pm 1.7$ | $24.2 \pm 1.2$ | $24.7 \pm 2.2$ |
| $M = 7$ | $24.7 \pm 1.3$ | $23.6 \pm 1.7$ | $26.7 \pm 1.2$ | $27.2 \pm 1.5$ | $28.0 \pm 2.1$ |

### 5.3. Synthetic Experiments

**Treatment Intervention.** In Section 4.2, we introduced the crucial estimands $N_t^\omega(F_H)$, which need to determine the intervention distribution of treatments over the $M$ periods from $t - M + 1$ to $t$. Let $\lambda_{Z_j}$ denote the intensity function of the spatial point process generating $Z_j$. We replace $\lambda_{Z_j}$ with the form: $h_j = c \times \log\left(\lambda_{Z_j}\right)$, $c$ is a constant, $j \in \{t - M + 1, t - M + 2, \ldots, t\}$, and $\log\left(\cdot\right)$ represents the natural logarithm. **The constant $c$ is introduced to control the magnitude of the intensity function.** A larger $c$ corresponds to the larger values of the intensity function, indicating an increased average density of treatments. Under our spatial-temporal setting, **parameter $M$ is crucial for regulating the duration of the intervention. A larger $M$ signifies an extended period of intervening in the treatments.** In synthetic experiments, $c \in \{3, 4, ..., 7\}$ and $M \in \{1, 3\}$. Details of treatment intervention are in Appendix G.

**Results.** We generate three synthetic datasets with time lengths of 32, 48, and 64 (i.e., $T = 32, 48, 64$.). We conduct experiments on each dataset, and in the main text, we present the experimental results of $M = 1$. Additional experiment results are in Appendix I. For $M = 1$, we calculate different $c \in \{3, 4, \ldots, 7\}$, gradually increasing the intensity of the treatment. The results are presented in Figure 3. Figure 3 visually illustrates our results. Across different time lengths, our method consistently outperforms the baselines.

### 5.4. Real Data Experiments

In real data experiments, we consider conflicts in Colombia as the treatment variable $Z$ and forest loss in Colombia as the outcome variable $Y$. We set $M \in \{1, 3, 5, 7\}$, and $c \in \{3, 4, ..., 7\}$, representing a gradual increase in the intensity and duration of conflicts.

**Results.** The results are illustrated in Table 1. Table 1 illustrates that the estimated forest loss increases as $M$ and $c$ increase. Thus, our results suggest that longer and more intense conflicts in Colombia may increase forest loss. Previous studies De Jong et al. (2007); Eniang et al. (2007); Garzón & Valánszki (2020); Kanyamibwa (1998) conclude that conflicts may harm natural resources like forests, which is consistent with our conclusion.

### 5.5. Ablation Studies

**Replace Transformer with RNNs.** In Section 4.6.3, we discuss the reason for choosing the Transformer as the backbone to estimate the intensity function. To further demonstrate the Transformer's superiority over RNNs, we replace the Transformer with RNNs and replicate the synthetic experiments. We present the results of T=64, M=1 in Figure 4, other results and details of RNNs are in Appendix J.1. According to Figure 4, results show that the estimation ability of RNNs is inferior to that of the Transformer.

**Relax the Poisson Assumption.** To validate the robustness of our methods over Assumption 2 (Poisson Assumption), we relax it and replicate the synthetic experiments. Specifically, we add the Gaussian kernel to the intensity functions of synthetic data, thus breaking the standard setting of the Poisson point process. We compare the results with the standard Poisson setting. We present the results of setting T=64, M=1 in Figure 5. The implementation details and other results are in Appendix J.2. According to Figure 5, relaxing the Poisson assumption does not lead to significant degradation in the estimation performance of our method, demonstrating its robustness to data distribution.

### 5.6. Computation Efficiency

Now we introduce the computing platform and computation efficiency. All experiments were conducted on an NVIDIA RTX 4090 GPU and an Intel Core i7 14700KF processor. In synthetic experiments, the synthetic data dimension could be (100,100,192). In addition, the time for one experimental setting was less than 10 minutes.

## 6. Conclusion

In this work, we study the estimation of counterfactual outcomes for spatial-temporal data and develop estimators with deep learning. Our experiments on synthetic datasets demonstrate the superior estimation capability of our estimator over four baselines, and experiments on real-world datasets provide a valuable conclusion to the causal effect of conflicts on forest loss in Colombia. For future work, we look to consider more general types of spatial-temporal data, moving beyond discrete point process data to encompass more general spatial-temporal stochastic processes.

## Acknowledgements

This work was partially supported by the National Natural Science Foundation of China (No. 62372459).

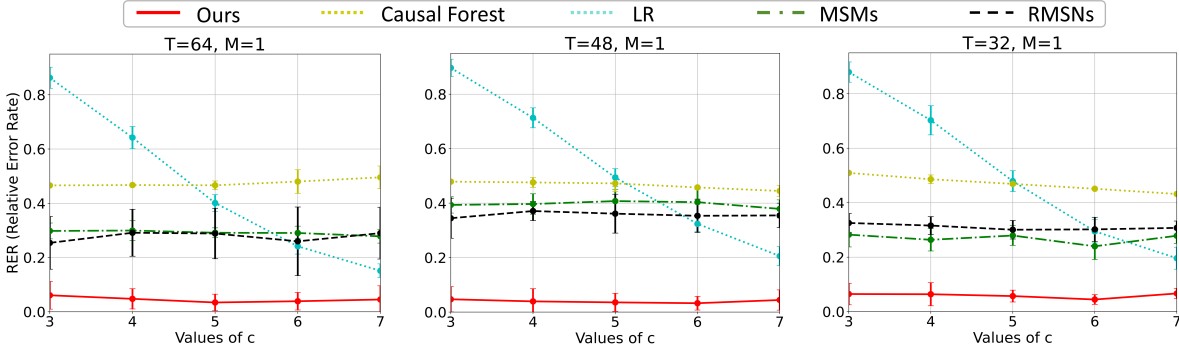

*Figure 3.* Synthetic experiments results of $M = 1$. The horizontal axis represents the values of $c$, while the vertical axis represents the relative error rate (RER). Lower lines in the graph correspond to methods with higher estimation accuracy.

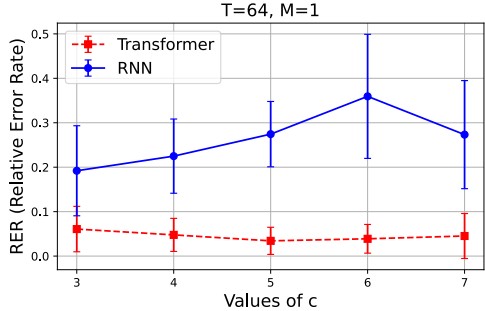

*Figure 4.* Comparison results of Transformer and RNN.

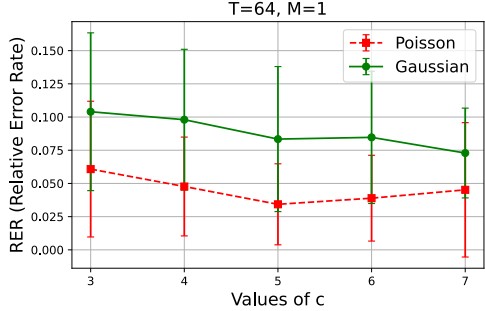

*Figure 5.* Results of relaxing the Poisson assumption.

## Impact Statement

In this work, we propose a framework to estimate the counterfactual outcomes with spatial-temporal attributes, which has positive societal impacts of helping predict the counterfactual outcomes of spatial-temporal data. Besides, all data used in this work are publicly available, and our newly released assets are the source code of our paper, which does not contain unsafe images or text.

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

# Supplementary Materials of Transformer-Based Spatial-Temporal Counterfactual Outcomes Estimation

## A. Notations

### A.1. Causal Model

Table 2. Notations for the causal model

| | |
|---|---|
| $\Omega$ | Whole spatial region |
| $\omega$ | One specific spatial region |
| $\gamma = \{1, 2, ..., T\}$ | Time index set |
| $s$ | Spatial location |
| $t$ | Time |
| $T$ | Number of time periods |
| $Z_t$ | Treatment in time $t$ |
| $Z_{\leq t}$ | Treatment up to time $t$ |
| $Z_{[t-M+1,t]}$ | Treatment between time $t - M + 1$ and time $t$ |
| $S_{z_t}$ | Treatment location in time $t$ |
| $Y_t(Z_{\leq t})$ | Potential outcome in time $t$ |
| $Y_t^{ob}(z_{\leq t})$ | Observed outcome in time $t$ |
| $S_{Y_t^{ob}(z_{\leq t})}$ | Outcome location in time $t$ |
| $X_t$ | Covariates in time $t$ |
| $x_t$ | Covariates realization in time $t$ |
| $h_{\leq t}$ | Observed historical information up to time $t$ |
| $H_{\leq t}$ | Potential historical information up to time $t$ |

### A.2. Intervention

Table 3. Notation for treatment intervention

| | |
|---|---|
| $h$ | Intensity function of spatial point process |
| $M$ | Duration of treatment intervention |
| $F_h$ | Distribution of spatial point patterns with intensity $h$ |
| $F_H$ | Joint distribution of $M$ independent $F_h$ |
| $F_h(z_t)$ | Distribution of $z_t$ is be assigned to $F_h$ |
| $z_{\leq t}(F_H)$ | Distribution of $z_{[t-M+1,t]}$ in $z_{\leq t}$ is be assigned to $F_H$ |

### A.3. Estimands

Table 4. Notation for interest estimands

| | |
|---|---|
| $N_t^{\omega}(F_H)$ | Expected number of outcomes that occur in time $t$ and region $\omega$ under intervention distribution $F_H$ |
| $N_{\omega}(F_H)$ | Average $N_t^{\omega}(F_H)$ over time $M$ to time $T$ |

## A.4. Estimators

*Table 5.* Notation for estimators

| | |
|---|---|
| $\hat{Y}_t(F_H, s)$ | Estimated intensity function of $y_t(z_{\leq t}(F_H))$ |
| $\hat{N}_t^\omega(F_H)$ | Estimator for $N_t^\omega(F_H)$ |
| $\hat{N}_\omega(F_H)$ | Estimator for $N_\omega(F_H)$ |
| $\lambda_{Y_t^{ob}(z_{\leq t})}(s)$ | Intensity function of spatial Poisson point process that generates $Y_t^{ob}(z_{\leq t}(F_H))$ |

# B. Running Example of the Estimands

Now we provide a running example for the estimands. For simplicity, consider the case of $t = 8$ and $M = 3$. Then the estimands

$$N_8^\omega(F_H) = \int_{Z^3} |S_{Y_8^{ob}(z_{\leq 8}(F_H))} \cap \omega| dF_H(z_{[6,8]}),$$

represent the expected number of outcomes in $t = 8$ and region $\omega$ under distribution $F_H$. Next, we employ the Table 6 to elaborate on each term of the estimands.

*Table 6.* Interpretation of key terms in the counterfactual estimands.

| Term | Interpretation |
|---|---|
| $t = 8$, $M = 3$ | Evaluates outcomes at time 8, considering 3-times intervention persistence (times 6–8). |
| $z_{[6,8]}$ | Sequence of treatment variables over the time window [6,8]. |
| $F_H(z_{[6,8]})$ | Joint probability distribution of $z_{[6,8]}$ under counterfactual intervention strategy $F_H$. |
| $\left| S_{Y_8^{ob}(z_{\leq 8}(F_H))} \cap \omega \right|$ | Observed outcome counts in region $\omega$ at time 8, under $F_H$. |
| $Z^3$ | All possible values of $z_{[6,8]}$. |
| $\int_{Z^3} \cdot \, dF_H(z_{[6,8]})$ | Expectation computation over all possible values of $z_{[6,8]}$. |

# C. Proofs

## C.1. Assumptions

**Assumption 1: Unconfoundedness.** Papadogeorgou et al. (2022) proposed the unconfoundedness assumption for spatial-temporal data, in which they assume that given $h_{\leq t}$, $Z_t$ is not dependent on any past or future potential outcomes and covariates. Their assumption is quite strict, as the future is unlikely to affect the past, so we relax their assumption and assume that condition on $h_{\leq t}$, $Z_t$ is not dependent on $H_{\leq t}$:

$$Z_t \perp\!\!\!\perp H_{\leq t} | h_{\leq t}.$$

**Assumption 2: Overlap.** For any $z_t \in Z_t$, $t \in \gamma$ and $h$, there exists a unique constant $\delta_z > 0$, such that $\frac{e_t(z_t)}{p_h(z_t)} > \delta_z$. This assumption ensures that all treatment point patterns in any intervention distribution $F_h$ are also possible in the observed world. Papadogeorgou et al. (2022) only use one treatment distribution $F_h$ in their estimators. In contrast, we used different treatment distributions in our estimators, so here we strictly assume a unique constant $\delta_z$ exists for any treatment distributions.

**Assumption 3.** Let $|\cdot|$ denote the number of elements of a set, and $Var$ denote the variance.

$(a)$ There exists a constant $\delta_Y > 0$ such that $|S_{Y_t^{ob}}(z_{\leq t})| < \delta_Y$ for all $t \in \gamma$ and $z_{\leq t}$.

$(b)$ Let $v_t = Var[\prod_{j=t-M+1}^{t} \frac{p_{h_j}(z_j)}{e_j(z_j)} N_\omega(Y_t) | H_{\leq t-M}]$ there exists a constant $v > 0$ such that $\frac{1}{T-M+1} \sum_{t=M}^{T} v_t \xrightarrow{p} v$ as $T \to \infty$.

$(c) \, |\int_\omega \lambda_{S_{Y_t^{ob}(z_{\leq t})}}(s)ds - N_\omega(Y_t)| < \beta$ and $\beta$ is an infinitesimals.

$(d) \, \beta$ is $o(\frac{1}{\sqrt{T}})$.

In Assumption 3, $(a)$ means that the number of outcome events at any time has an upper bound under any treatment. In our real-world scenario, we consider forest loss as the outcome event. Forest loss cannot be infinite, so $(a)$ is reasonable. $(b)$ assumes the convergence in probability of a sequence. In $(c)$, $N_\omega(Y_t)$ is the actual number of outcome events in time $t$ and region $\omega$ because our neural network smoothing can be viewed as an estimate of the intensity function of a point process. We design it so that the output values of the neural network are as large as possible where the outcome occurs, and as small as possible where the outcome does not occur, so $(c)$ is reasonable. In $(d)$, we assume that $\beta$ tends to zero at a faster rate than $\frac{1}{\sqrt{T}}$, i.e., $\beta\sqrt{T} \to 0$ as $T \to \infty$.

## C.2. Propositions

**Proposition 1.** The propensity score is a balancing score. For any $t \in \gamma$,

$$Z_t \perp\!\!\!\perp h_{\leq t}|e_t(z_t).$$

**Proposition 2.** Dimensional reduction property of the propensity score: if $Z_t \perp\!\!\!\perp H_{\leq t}|h_{\leq t}$ then $Z_t \perp\!\!\!\perp H_{\leq t}|e_t(z_t)$.

**Proposition 3: The consistency and asymptotic normality of the estimator.** Let $Var$ denote the Variance, $N_\omega(Y_t)$ denote the number of the observed outcome $Y_t$ in region $\omega$. If all assumptions hold, and $T \to \infty$, we have that

$$\sqrt{T}(\hat{N}_\omega(F_H) - N_\omega(F_H)) \xrightarrow{d} N(0, v),$$

where $v = lim_{T\to\infty} \frac{1}{T-M+1} \sum_{t=M}^T v_t$ and $v_t = Var[\prod_{j=t-M+1}^t \frac{p_{h_j}(z_j)}{e_j(z_j)} N_\omega(Y_t)|H_{\leq t-M}]$.

## C.3. Definition

**Definition: Martingale Difference Series (Van der Vaart, 2010).** Let $(\Omega, \mathcal{F}, \Pr)$ be a probability space, a filtration $\mathcal{F}_t = \{\mathcal{F}_t; t \geq 0\}$ is a non-decreasing collection of $\sigma-field$ on $\mathcal{F}$, (e.g. $\mathcal{F}_0 \subset \mathcal{F}_1 \subset ... \subset \mathcal{F}_t \subset ... \subset \mathcal{F}$). Let $X_t = \{X_t; t \geq 0\}$ be a time series. A martingale difference series relative to a given filtration is a time series $X_t$ such that, for any $t$:

(1) $X_t$ is $\mathcal{F}_t$ measurable.

(2) $E[|X_t|] < \infty$.

(3) $E[X_t|\mathcal{F}_{t-1}] = 0$.

## C.4. Theorem

**Theorem: Central limit theorem for Martingale difference series (Van der Vaart, 2010).** If $X_t$ is a martingale difference series relative to the filtration $\mathcal{F}_t$, and there exists a constant $v > 0$, such that $\frac{1}{n}\sum_{t=1}^n E[X_t^2|\mathcal{F}_{t-1}] \xrightarrow{p} v$, and such that $\frac{1}{n}\sum_{t=1}^n E[X_t^2 I_{|X_t|>\epsilon\sqrt{n}}|\mathcal{F}_{t-1}] \xrightarrow{p} 0$ for any $\epsilon > 0$, then $\sqrt{n}\overline{X_n} \xrightarrow{d} N(0, v)$.

## C.5. Proofs for Propositions

**Proofs for Proposition 1.** We need to show that $\mathbb{P}(Z_t = z_t|e_t(z_t), h_{\leq t}) = \mathbb{P}(Z_t = z_t|e_t(z_t))$. Since $e_t(z_t)$ is a function of $h_{\leq t}$ then $\mathbb{P}(Z_t = z_t|e_t(z_t), h_{\leq t}) = \mathbb{P}(Z_t = z_t|h_{\leq t}) = e_t(z_t)$,

$$\mathbb{P}(Z_t = z_t|e_t(z_t)) = E[\mathbb{P}(Z_t = z_t|h_{\leq t})|e_t(z_t)] = E[e_t(z_t)|e_t(z_t)] = e_t(z_t).$$

Based on the above, we prove the Proposition 1.

**Proofs for Proposition 2.**   We need to show that $\mathbb{P}(Z_t = z_t | H_{\leq t}, e_t(z_t)) = \mathbb{P}(Z_t = z_t | e_t(z_t))$. Since $e_t(z_t)$ is a function of $h_{\leq t}$ and $h_{\leq t} \subset H_{\leq t}$, then $\mathbb{P}(Z_t = z_t | H_{\leq t}, e_t(z_t)) = \mathbb{P}(Z_t = z_t | H_{\leq t})$ and

$$
\begin{aligned}
\mathbb{P}(Z_t = z_t | H_{\leq t}) &= \mathbb{P}(Z_t = z_t | H_{\leq t}, h_{\leq t}) \\
&= \mathbb{P}(Z_t = z_t | h_{\leq t}) \quad \text{(Unconfoundedness)} \\
&= e_t(z_t) \\
&= \mathbb{P}(Z_t = z_t | e_t(z_t)).
\end{aligned}
$$

Based on the above, we prove the Proposition 2.

**Proofs for Proposition 3.**   Let $Err_t = \hat{N}_t^\omega(F_H) - N_t^\omega(F_H)$ represent the estimation error at time $t$. We divide the estimation error $Err_t$ into two parts: the first part is $E_{1t}$, which comes from the treatment allocation, and the second part is $E_{2t}$, which comes from the spatial smoothing of our neural network. To be specific,

$$
Err_t = \prod_{j=t-M+1}^{t} \frac{p_{h_j}(z_j)}{e_j(z_j)} \int_\omega \lambda_{S_{Y_t^{ob}(z_{\leq t})}}(s)ds - N_t^\omega(F_H) = E_{1t} + E_{2t}
$$

where

$$
E_{1t} = \prod_{j=t-M+1}^{t} \frac{p_{h_j}(z_j)}{e_j(z_j)} N_\omega(Y_t) - N_t^\omega(F_H)
$$

and

$$
E_{2t} = \prod_{j=t-M+1}^{t} \frac{p_{h_j}(z_j)}{e_j(z_j)} \int_\omega \lambda_{S_{Y_t^{ob}(z_{\leq t})}}(s)ds - N_\omega(Y_t).
$$

We will show that,

$(i)$ $\sqrt{T}\left(\frac{1}{T-M+1} \sum_{t=M}^{T} E_{1t}\right) \xrightarrow{d} N(0, v)$,

$(ii)$ $\sqrt{T}\left(\frac{1}{T-M+1} \sum_{t=M}^{T} E_{2t}\right) \to 0$.

**Proof of the asymptotic normality of the first part of the estimation error.**   We use Definition C.3, the definition of the Martingale difference series, and Theorem C.4, the central limit theorem for the Martingale difference series to prove $(i)$ $\sqrt{T}\left(\frac{1}{T-M+1} \sum_{t=M}^{T} E_{1t}\right) \xrightarrow{d} N(0, v)$.

**Lemma 1.** $E_{1t}$ is a martingale difference series with respect to the filtration $\mathcal{F}_t = H_{\leq t-M+1}$.

**Proof for Lemma 1:** We need to show that,

(1) $E_{1t}$ is $H_{\leq t-M+1}$ measurable,

(2) $E[|E_{1t}|] < \infty$,

(3) $E[E_{1t} | \mathcal{F}_{t-1}] = E[E_{1t} | H_{\leq t-M}] = 0$.

It is easy to prove that $(1)$ holds from the definitions of $E_{1t}$ and $H_{\leq t-M+1}$. From Assumption 2 and Assumption 3 $(c)$ we have

$$
|E_{1t}| \leq \left| \prod_{j=t-M+1}^{t} \frac{p_{h_j}(z_j)}{e_j(z_j)} N_\omega(Y_t) \right| + |N_t^\omega(F_H)| < \delta_Y(1 + \delta_z^{-M}) < \infty.
$$

Therefore, $E[|E_{1t}|] < \delta_Y(1 + \delta_z^{-M}) < \infty$, $(2)$ is proved.

For the proof of $(3)$, we will show that

$$
E\left[ \prod_{j=t-M+1}^{t} \frac{p_{h_j}(z_j)}{e_j(z_j)} N_\omega(Y_t) | H_{\leq t-M} \right] = N_t^\omega(F_H),
$$

$$
\begin{aligned}
E\Big[\prod_{j=t-M+1}^{t}\frac{p_{h_j}(z_j)}{e_j(z_j)}N_\omega(Y_t)|H_{\leq t-M}\Big] &= \int \prod_{j=t-M+1}^{t}\frac{p_{h_j}(z_j)}{e_j(z_j)}N_\omega(Y_t^{ob}(z_{\leq t}(F_H))) \times \mathbb{P}(z_{t-M+1}|H_{\leq t-M}) \\
&\quad \times \mathbb{P}(z_{t-M+2}|H_{\leq t-M}, z_{t-M+1}) \times ... \times \mathbb{P}(z_t|H_{\leq t-M}, z_{[t-M+1,t-1]})dz_{[t-M+1,t]} \\
&= \int \prod_{j=t-M+1}^{t}\frac{p_{h_j}(z_j)}{e_j(z_j)}N_\omega(Y_t^{ob}(z_{\leq t}(F_H))) \times \mathbb{P}(z_{t-M+1}|H_{\leq t-M}) \\
&\quad \times \mathbb{P}(z_{t-M+2}|H_{\leq t-M+1}) \times ... \times \mathbb{P}(z_t|H_{\leq t-1})dz_{[t-M+1,t]} \\
&= \int \prod_{j=t-M+1}^{t} p_{h_j}(z_j)N_\omega(Y_t^{ob}(z_{\leq t}(F_H)))dz_{[t-M+1,t]} \quad \text{(Unconfoundedness)} \\
&= N_t^\omega(F_H).
\end{aligned}
$$

Then we have

$$
\begin{aligned}
E[E_{1t}|H_{\leq t-M}] &= E\Big[\prod_{j=t-M+1}^{t}\frac{p_{h_j}(z_j)}{e_j(z_j)}N_\omega(Y_t)|H_{\leq t-M}\Big] - E[N_t^\omega(F_H)|H_{\leq t-M}] \\
&= N_t^\omega(F_H) - N_t^\omega(F_H) \\
&= 0.
\end{aligned}
$$

Based on the above, we prove the Lemma 1.

**Lemma 2.** $\frac{1}{T-M+1}\sum_{t=M}^{T}E[E_{1t}^2 I(|E_{1t}| > \epsilon\sqrt{T-M+1}|\mathcal{F}_{t-1})] \xrightarrow{p} 0$, for any $\epsilon > 0$.

**Proof for Lemma 2.** We will show $I(|E_{1t}| > \epsilon\sqrt{T-M+1}|\mathcal{F}_{t-1}) \to 0$ as $T \to \infty$. From Assumption 2 and Assumption 3 $(b)$, we can obtain $|E_{1t}| < \delta_Y(1 + \delta_z^{-M})$. We show $\delta_Y(1 + \delta_z^{-M}) \leq \epsilon\sqrt{T-M+1}$, as $T \to \infty$,

$$
\begin{aligned}
\epsilon^{-1}\delta_Y(1 + \delta_z^{-M}) &\leq \sqrt{T-M+1} \\
[\epsilon^{-1}\delta_Y(1 + \delta_z^{-M})]^2 &\leq T-M+1 \\
M - 1 + [\epsilon^{-1}\delta_Y(1 + \delta_z^{-M})]^2 &\leq T,
\end{aligned}
$$

let $T_0 = \lceil M - 1 + [\epsilon^{-1}\delta_Y(1 + \delta_z^{-M})]^2 \rceil$, when $T \geq T_0$, we have $\delta_Y(1 + \delta_z^{-M}) \leq \epsilon\sqrt{T-M+1}$ and $|E_{1t}| < \epsilon\sqrt{T-M+1}$. Therefore, we have $I(|E_{1t}| > \epsilon\sqrt{T-M+1}|\mathcal{F}_{t-1}) \to 0$ as $T \to \infty$. Based on the above, we prove the Lemma 2.

We have $E[E_{1t}^2|\mathcal{F}_{t-1}] = Var[E_{1t}|\mathcal{F}_{t-1}] = Var[\prod_{j=t-M+1}^{t}\frac{p_{h_j}(z_j)}{e_j(z_j)}N_\omega(Y_t)|H_{\leq t-M}]$, from Assumption 3$(b)$, we have

$$
\frac{1}{T-M+1}\sum_{t=M}^{T}E[E_{1t}^2|\mathcal{F}_{t-1}] \xrightarrow{p} v.
$$

Combining Lemma 1, Lemma 2, and the Central limit theorem for the Martingale difference series C.4, we have

$$
\sqrt{T}\Big(\frac{1}{T-M+1}\sum_{t=M}^{T}E_{1t}\Big) \xrightarrow{d} N(0, v).
$$

Based on the above, we prove the asymptotic normality of the first part of the estimation error.

**Proof of the convergence in probability of the second part of the estimation error to zero.** The second part of the estimation error represents the difference between the integral of the outcome of the neural network smoothing and the actual number of outcomes. We will show

$$
\sqrt{T}\Big(\frac{1}{T-M+1}\sum_{t=M}^{T}E_{2t}\Big) \to 0.
$$

Let $\alpha_t = \prod_{j=t-M+1}^{t} \frac{p_{h_j}(z_j)}{e_j(z_j)}$, then

$$\left| \frac{1}{T-M+1} \sum_{t=M}^{T} E_{2t} \right| = \left| \frac{1}{T-M+1} \sum_{t=M}^{T} \alpha_t \left[ \int_\omega \lambda_{S_{Y_t^{ob}(z_{\leq t})}}(s)ds - N_\omega(Y_t) \right] \right|.$$

From Assumption 2, we obtain $\alpha_t < \delta_z^{-M}$. From Assumption 3$(c)$, we obtain $|\int_\omega \lambda_{S_{Y_t^{ob}(z_{\leq t})}}(s)ds - N_\omega(Y_t)| < \beta$. Then we have

$$\left| \frac{1}{T-M+1} \sum_{t=M}^{T} E_{2t} \right| < \delta_z^{-M} \left| \frac{1}{T-M+1} \sum_{t=M}^{T} \left[ \int_\omega \lambda_{S_{Y_t^{ob}(z_{\leq t})}}(s)ds - N_\omega(Y_t) \right] \right|$$

$$\leq \delta_z^{-M} \frac{1}{T-M+1} \sum_{t=M}^{T} \left| \int_\omega \lambda_{S_{Y_t^{ob}(z_{\leq t})}}(s)ds - N_\omega(Y_t) \right|$$

$$< \delta_z^{-M} \frac{1}{T-M+1} (T-M+1)\beta$$

$$= \delta_z^{-M} \beta.$$

Because $\beta$ can be arbitrarily small so $\delta_z^{-M}\beta$ can also be arbitrarily small, combined with Assumption 3 $(d)$, we conclude that $\sqrt{T}(\frac{1}{T-M+1}\sum_{t=M}^{T} E_{2t}) \to 0$.

Combining the proof of the asymptotic normality of the first part of the estimation error and the proof of the convergence to 0 of the second part of the estimation error, we prove $\sqrt{T}(\hat{N}_\omega(F_H) - N_\omega(F_H)) \xrightarrow{d} N(0, v)$ as $T \to \infty$.

## D. Identifiability Proof

This section provides proof of identifiability for the estimands. Under the potential outcomes framework, identifiability refers to whether an estimand can be represented using observable data. The formal definition is given below:

**Definition D.1. (Identifiability)** A parameter $\theta$ is said to be identifiable if it can be expressed as a function of the distribution of observed data under certain assumptions. The parameter $\theta$ is said to be nonparametrically identifiable if it can be expressed as such a function without any parametric assumptions on the model.

In the above definition, the parameter $\theta$ refers to the estimation target, which could be a causal effect or a counterfactual outcome. Identifiability is crucial in observational studies because an estimand must be identifiable in order to be estimable from observational data. If the estimation target depends on unobservable data, then it is non-identifiable and cannot be estimated. To derive the identifiability of the estimation target in this work, we introduce the following reasonable assumptions:

**Assumption (Ignorability).** We assume that, given the observed history $h_{\leq t}$, the treatment variable $Z_t$ is independent of the latent history $H_{\leq t}$:

$$Z_t \perp\!\!\!\perp H_{\leq t} \mid h_{\leq t}. \tag{11}$$

Note that Papadogeorgou et al. (2022) assume that $Z_t$ is independent of all past and future potential outcomes and covariates given $h_{\leq t}$. Their assumption is overly strict, as the future cannot causally affect the past (due to the unidirectional nature of causality in time). Therefore, we relax their assumption by only requiring $Z_t$ to be independent of the latent history $H_{\leq t}$.

**Assumption (Poisson Assumption).** We assume that $Z_t$, $Y_t(Z_{\leq t})$, and $Z_t \mid h_{\leq t-1}$ are generated by spatial Poisson point processes. This assumption is reasonable because spatial Poisson point processes are widely used statistical tools to model the distribution of discrete events over spatial regions (Cressie & Wikle, 2015).

**Assumption (Consistency).** At any time and in any region, if a treatment sequence $z_{\leq t}$ is assigned, then the observed outcome $Y_t^{ob}$ must equal the potential outcome under that treatment, i.e.,

$$Y_t^{ob} = Y_t(Z_{\leq t} = z_{\leq t}). \tag{12}$$

This assumption ensures that the observed outcomes are consistent with the potential outcomes, enabling the estimation of counterfactual outcomes and causal effects.

**Theorem D.2** (Identifiability of the Estimation Target). *If Assumptions Ignorability, Poisson, and Consistency hold, then the estimands $N_t^\omega(F_H)$ is identifiable.*

*Proof.* By the definition of $N_t^\omega(F_H)$, it suffices to show that $E[Y_t^{ob}(z_{\leq t}(F_H))]$ is identifiable.

$$E[Y_t^{ob}(z_{\leq t}(F_H))] = E[E[Y_t^{ob}(z_{\leq t}(F_H)) \mid h_{\leq t}]] \tag{13}$$

$$= E[E[Y_t^{ob}(z_{\leq t}(F_H)) \mid h_{\leq t}, z_{\leq t} \sim F_H]] \tag{14}$$

$$= E[E[Y_t(Z_{\leq t} \sim F_H) \mid h_{\leq t}, z_{\leq t} \sim F_H]] \tag{15}$$

$$= E[E[Y_t^{ob} \mid h_{\leq t}, z_{\leq t} \sim F_H]], \tag{16}$$

Equation (13) follows from the tower property of conditional expectation, (14) is derived using the ignorability assumption (since $Y_t^{ob}(z_{\leq t}(F_H)) \in H_{\leq t}$), and both (15) and (16) follow from the consistency assumption. The Poisson assumption ensures that the distributions of the potential outcomes and treatments are known, further reinforcing the identifiability of the estimation target. □

The above establishes the identifiability of the estimands, which guarantees the solvability of the proposed estimation problem.

## E. Derivation of Property of Poisson Point Process

Let $\lambda(s)$ denote the intensity function of a Poisson point process, $A$ denote a spatial region, $\mathcal{N}(A)$ denote the number of events occurring within region $A$, and $E[\mathcal{N}(A)]$ denote the expected number of events occurring within region $A$.

According to the property of the Poisson point process, we have the following:

$$\mathcal{N}(A) \sim \text{Poisson}(\lambda_A),$$

where $\text{Poisson}(\cdot)$ denotes the Poisson distribution and $\lambda_A$ is its parameter.

Then

$$E[\mathcal{N}(A)] = \sum_{i=0}^{\infty} i\mathbb{P}(\mathcal{N}(A) = i) = \sum_{i=1}^{\infty} i\frac{e^{-\lambda_A}\lambda_A^i}{i!} = \lambda_A e^{-\lambda_A} \sum_{i=1}^{\infty} \frac{\lambda_A^{i-1}}{(i-1)!} = \lambda_A e^{-\lambda_A} e^{\lambda_A} = \lambda_A.$$

According to the definition of the intensity function we have:

$$E[\mathcal{N}(A)] = \int_A \lambda(s)ds = \lambda_A.$$

## F. Derivation of the Training Objective Function

The objective function Eq. (10) is mainly derived from the likelihood function of the Poisson point process and the objective function of the variational autoencoder (VAE). Specifically, we employ the idea of maximum likelihood estimation (MLE), and the likelihood function of the Poisson point process is shown below:

$$\sum_i log(\lambda^*(s_i)) - \int_S \lambda^*(u)du.$$

MLE for the intensity function of the Poisson point process seeks the optimal intensity function $\lambda^*(\cdot)$ from the data that optimizes the above function. As for the $KL(q\|p)$ in Eq. (10), it's a common component of VAE. Therefore, our method can be seen as a combination of MLE and VAE. We use a neural network as the intensity function and seek the optimal intensity function by optimizing the objective function Eq. (10).

## G. Explanation of the Treatment Intervention

In this section, we provide a brief explanation of why the intervention treatment duration $M$ has an impact on the results. In our problem setting, we assume treatments have delayed effects, i.e., temporal carryover. For example, an educational program results in poorer performances at first, as students adjust to new learning methods, but eventually, they may get long-term improvements. Therefore, under our spatial-temporal setting, the larger $M$ may influence the estimation.

# H. Explanation of the term "Counterfactual Probability"

In this section, we provide a brief explanation of why the counterfactual probability is "counterfactual". Recall that $p_h(z_t)$ is the probability density function of intervention distribution $F_h(\cdot)$. In this work, we aim to estimate the number of potential outcomes when the treatment follows an intervention distribution. We specify this intervention distribution, which does not necessarily exist in the observable data. For example, in the observable data, treatment may follow a distribution $P_A$, and we want to find out what will happen to outcomes when treatment follows another distribution $P_B$. Therefore, the counterfactual probability $p_h(z_t)$ is counterfactual and it's counter to the observable data or real-world.

# I. Additional Experiments and Their Results

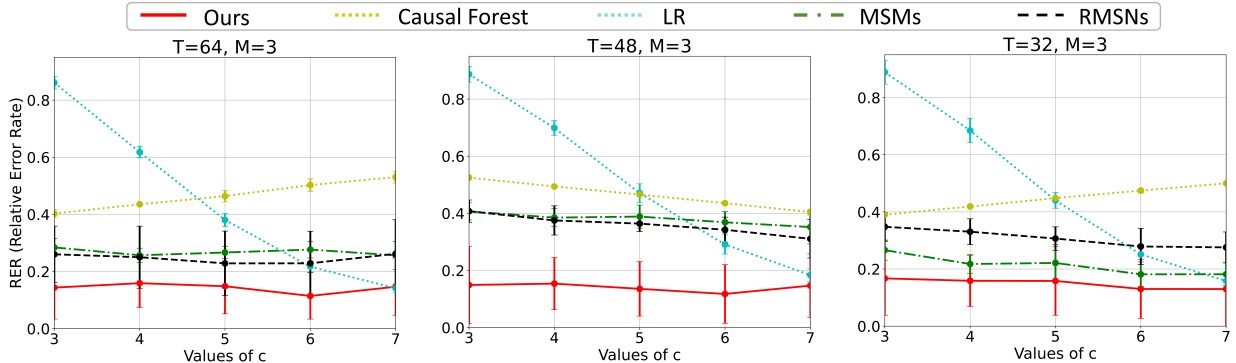

*Figure 6.* Experiments results of $M = 3$. The horizontal axis represents the values of $c$, while the vertical axis represents the relative error rate (RER). The lower lines in the graph correspond to methods with higher estimation accuracy. From left to right, the three columns respectively represent the experimental results with time lengths of 64, 48, and 32 ($T = 64, 48, 32$).

According to Figure 6, in the different experiment settings of M=3, the estimation errors of our method are relatively low in most cases, indicating the robustness of our method concerning time lengths and intervention settings.

# J. Details of the Ablation Studies

## J.1. Details of the RNNs

### J.1.1. PARAMETERS OF THE RNNS

We employ the Gated Recurrent Unit (GRU) network. The input_size = 32, hidden_size = 32, num_layers = 3, dropout=0.1, batch_first = True.

### J.1.2. COMPARISON RESULTS OF RNNS

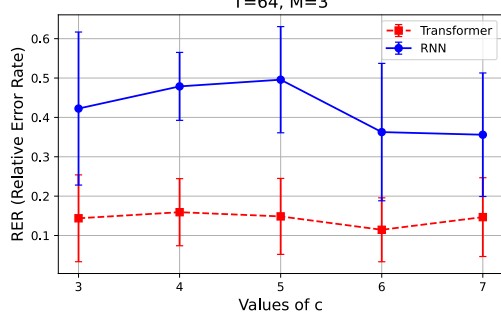
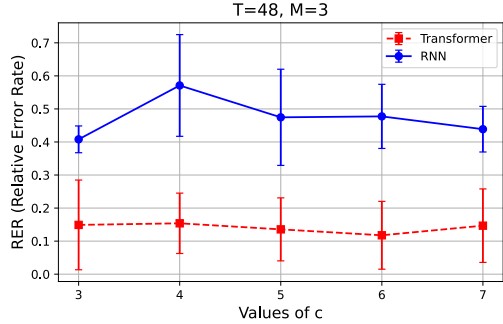

(a) Comparison results of T=64, M=3.        (b) Comparison results of T=48, M=3.

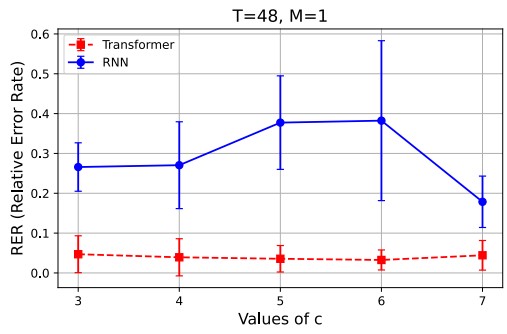

(a) Comparison results of T=48, M=1.

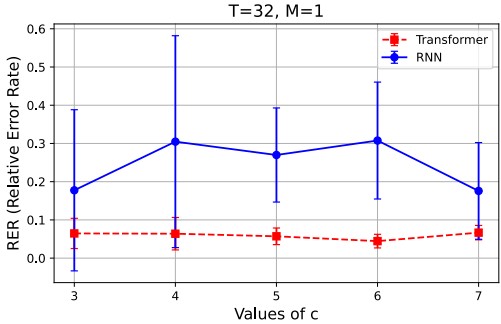

(b) Comparison results of T=32, M=1.

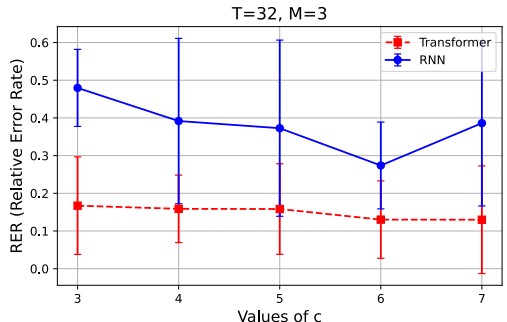

*Figure 9.* Comparison results of T=32, M=3.

According to the above comparison results with RNNs, the Transformer backbone is superior to the RNNs in most settings.

## J.2. Details of the Relaxation of Poisson Assumption

### J.2.1. SETTINGS OF INTENSITY FUNCTIONS

To relax the Poisson assumption, we add the Gaussian kernels to the intensity functions (see Appendix M.1) that generate the synthetic data. Specifically, we replace the $Z^*_{t-1}(s) = e^{-2D_{Z_{t-1}}(s)}$ with Gaussian kernel of $e^{-\frac{D^2_{Z_{t-1}}(s)}{2\sigma^2}}$, $Y^*_{t-1}(s) = e^{-2D_{Y_{t-1}}(s)}$ with $e^{-\frac{D^2_{Y_{t-1}}(s)}{2\sigma^2}}$, and $Z^*_{[t-3,t]}(s) = e^{-2D_{Z_{[t-3,t]}}(s)}$ with $e^{-\frac{D^2_{Z_{[t-3,t]}}(s)}{2\sigma^2}}$. The $\sigma$ is set to a constant of $\frac{1}{\sqrt{2}}$.

### J.2.2. COMPARISON RESULTS

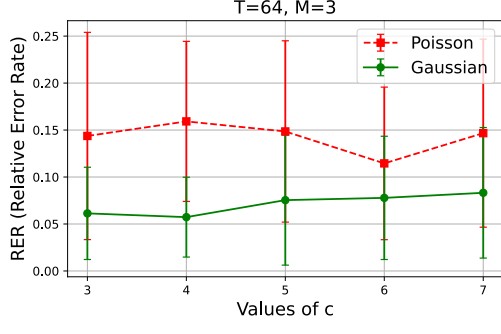

(a) Relaxing results of T=64, M=3.

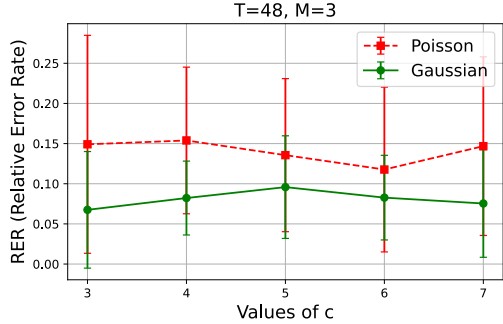

(b) Relaxing results of T=48, M=3.

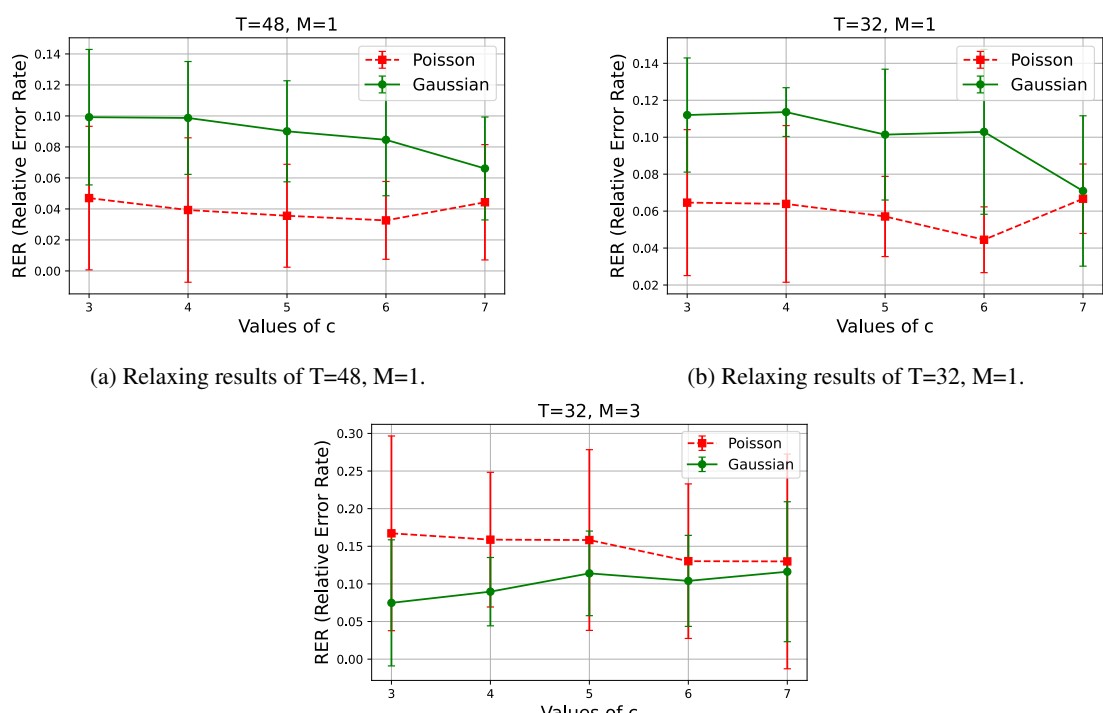

(a) Relaxing results of T=48, M=1.                    (b) Relaxing results of T=32, M=1.

*Figure 12.* Relaxing results of T=32, M=3.

## K. The Design Details of Convolutional Neural Networks

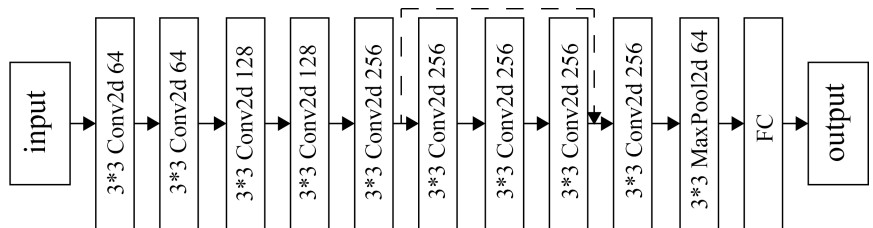

*Figure 13.* The design details of Convolutional Neural Networks.

## L. Details of the Real Dataset

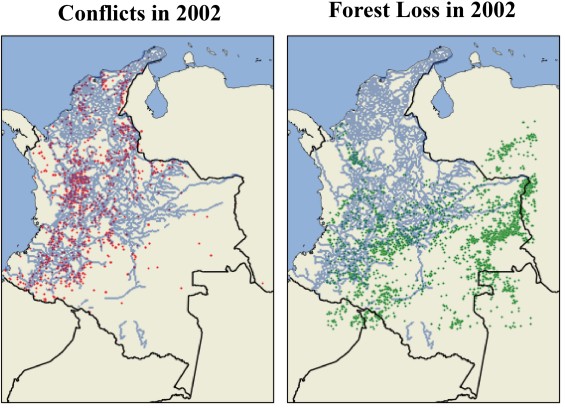

*Figure 14.* An example of the real dataset.

An example of the real dataset is shown in Figure 14. In Figure 14, the blue lines represent the roads in Colombia. The left part of the figure shows the conflicts in 2002 in Colombia, the red dots represent the conflict locations. The right part of the figure demonstrates the forest loss in 2002 in Colombia, the green dots denote the forest loss locations.

### L.1. Forest Change Data.

We only consider the "Year of gross forest cover loss event (loss year)" section of the Global Forest Change dataset, which is a raster data matrix (.tif file). The matrix elements are 0, representing no forest loss events, and values from 1 to 22, representing forest loss events occurring from 2001 to 2022 (Hansen et al., 2013). We selected a portion from the dataset that occurred within the territory of Colombia. We read the raster data matrix's elements from 2 to 22, calculating their latitude and longitude positions. As a result, we obtained the forest loss events that occurred in Colombia from 2002 to 2022. Data is publicly available online from `https://glad.earthengine.app/view/global-forest-change`.

### L.2. UCDP Georeferenced Event Dataset.

UCDP Georeferenced Event Dataset exists as tabular data (.xlsx file) (Croicu & Sundberg, 2015). We only consider the years in which conflicts occurred, the latitude and longitude coordinates of the conflict locations, and the country or region where the conflicts occurred, as contained in the tabular data. We directly filtered conflict events within Colombia from the tabular data, spanning from 2002 to 2022. Data is publicly available online from `https://ucdp.uu.se/downloads/index.html#ged_global`.

### L.3. Road Data.

We select several major roads within Colombia from Google Maps. Specifically, for convenience, we choose key points from the main roads of Colombia and connect these points to form a polyline, representing the road. The coordinates of selected key points are shown below:

- Road1: (-77.66, 0.77), (-76.42, 3.09), (-76.06, 4.29).

- Road2: (-76.06, 4.29), (-74.03, 4.66), (-73.04, 7.05), (-72.38, 7.75).

- Road3: (-76.02, 4.31), (-75.54, 6.15), (-73.11, 7.09), (-74.22, 10.97), (-72.23, 11.33).

- Road4: (-75.52, 6.15), (-74.69, 10.93).

- Road5: (-75.52, 6.22), (-76.77, 8.42).

The first part of the coordinate is the longitude, and the second part is the latitude.

## M. Synthetic Data Generating Process

We utilize reject sampling (Lavancier et al., 2015) to sample spatial point patterns of treatments and outcomes based on their intensity functions. Below, we introduce the setting for intensity functions.

### M.1. Intensity Functions Setting

We consider four covariates $X_1(s)$ and $X_2(s)$ and $X_3(s)$ and $X_4(s)$. $X_1(s) = e^{-3D_1(s)} + log(D_2(s))$, $X_2(s) = e^{-3D_3(s)}$. $D_1(s)$ is the distance from the location $s$ to the road on spatial area. $D_2(s)$ is the distance from $s$ to the border of the spatial area. $D_3(s)$ is the distance from $s$ to the center of the spatial area. $\lambda^{X_3}(s) = e^{a_0^3 + a_1^3 X_1(s)}$ and $\lambda^{X_4}(s) = e^{a_0^4 + a_1^4 X_1(s)}$ are two intensity function, we use the point patterns generated by them to create $X^3(s)$ and $X^4(s)$. Specifically, $a_0^3, a_1^3, a_0^4$ and $a_1^4$, are bias constant. Let $D_3(s)$ represent the distance from $s$ to the closet points generated by $\lambda^{X_3}(s)$, $D_4(s)$ represent the distance from $s$ to the closet points generated by $\lambda^{X_4}(s)$. Then $X_3(s) = e^{D_3(s)}$, $X_4(s) = e^{D_4(s)}$.

Based on all covariates, we determine the intensity function of the Poisson point process that generates treatment and outcome. Let $X(s) = (X_1(s), X_2(s), X_3(s), X_4(s))$ denote the covariates vector, and $Z_{t-1}^*(s) = e^{-2D_{Z_{t-1}}(s)}$, $Y_{t-1}^*(s) = e^{-2D_{Y_{t-1}}(s)}$, $D_{Z_{t-1}}(s)$ is the distance from $s$ to the closet point in $S_{Z_{t-1}}$, $D_{Y_{t-1}}(s)$ is the distance from $s$ to the closet point in $S_{Y_{t-1}}$. The

intensity function for the Poisson point process that generates $Z_t$ is shown as follows:

$$\lambda_{Z_t}(s) = e^{\beta_0 + \beta_X X(s) + \beta_Z Z^*_{t-1}(s) + \beta_Y Y^*_{t-1}(s)}. \tag{17}$$

In Eq. (17), $\beta_0, \beta_X, \beta_Z, \beta_Y$, are constant parameters. Let $Z^*_{[t-3,t]}(s) = e^{-2D_{Z_{[t-3,t]}}(s)}$, $D_{Z_{[t-3,t]}}(s)$ is the distance from $s$ to the closet point in $\bigcup_{j=t-3}^{t} S_{Z_j}$. The intensity function for the Poisson point process that generates $Y_t$ is shown as follows:

$$\lambda_{Y_t}(s) = e^{\gamma_0 + \gamma_X X(s) + \gamma_Z Z^*_{[t-3,t]}(s) + \gamma_Y Y^*_{t-1}(s)}. \tag{18}$$

In Eq. (18), $\gamma_0, \gamma_X, \gamma_Z, \gamma_Y$, are also constant parameters. In the synthetic data generating process, we set $\beta_0 = -1$, $\beta_X = (1,1,1,1)$, $\beta_Z = 1$, $\beta_Y = 1$, and $\gamma_0 = 1$, $\gamma_X = (1,1,1,1)$, $\gamma_Z = 1$, $\gamma_Y = 1$, $a_0^3 = -0.2$, $a_1^3 = 2.3$, $a_0^4 = -0.2$ and $a_1^4 = 2.8$.

## M.2. Ground Truth Generation

For the computation of the true counterfactual outcomes, we first employ the method described in Appendix M.1 to calculate the intensity function of the spatial Poisson point process generating $Y_t^{ob}(z_{\leq t}(F_H))$. Subsequently, we utilize this intensity function to generate samples of $Y_t^{ob}(z_{\leq t}(F_H))$. Finally, the average of samples is computed as the true $N_t^\omega(F_H)$.

## M.3. Synthetic Data Region

In synthetic data, the entire area is a rectangle with a length of 1 and a width of 1. For ease of computer processing, we divide this area into 100 squares. We draw some lines on the rectangle, which we consider as roads. The synthetic data area is shown in Figure 15. In Figure 15, the red line represents the straight road, while the green dashed line represents the curved road. Based on the synthetic data region, we calculate the intensity function of the treatments and outcomes. While it is possible to generate an arbitrary number of intensity functions, due to computational limitations, we have generated 32, 48, and 64 intensity functions for the treatments and outcomes, corresponding to time lengths of 32, 48, and 64. Afterward, based on the intensity functions, we use rejection sampling to generate point patterns of treatments and outcomes. To be specific, twenty spatial point patterns are generated from each intensity function.

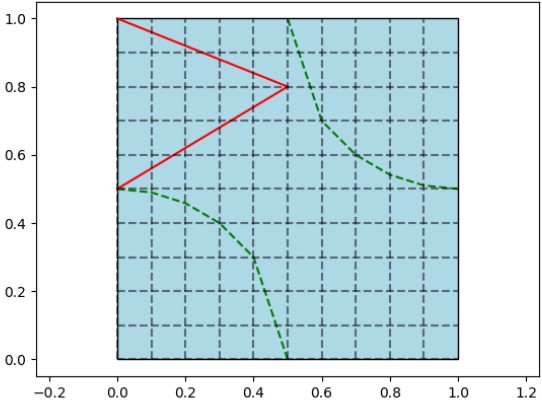

*Figure 15.* The synthetic spatial area. The red lines and green lines in the figure represent the synthetic roads.

# N. Details of Baselines

## N.1. Baselines Adaption

Now we introduce how we adapt baselines to our setting. Since baselines cannot directly handle high-dimensional data such as series of spatial point patterns, we transform all treatments ($z_t$) and outcomes ($Y_t$) into the number of events contained in the treatments and outcomes ($R(z_t)$ and $R(Y_t)$), $R()$ is the dimension reduction map defined in Eq. (5) in the main text.

Therefore, the data used for baselines are all scalar series. For baselines like MSMs, RMSNs, and Linear Regression (LR), we fit scalar outcomes on treatments and covariates and make a comparison with our method. For Causal Forest, we first use a linear regression model to fit scalar treatments on outcomes, and then train the causal forest model to estimate the treatment effects. Finally, we combine the causal forest and regression model to build the counterfactual outcomes.

### N.2. Marginal Structural Models (MSMs)

Marginal Structural Models (MSMs) (Robins et al., 2000) is the statistical tool used in observational studies to estimate causal effects, especially when dealing with time-varying exposures and confounders. MSMs employ the standard regression method as a base estimation model and adjust for these complexities using methods like inverse probability weighting (IPW) or g-estimation. In our configurations, we employ MSMs to fit outcomes on treatments and covariates and make a comparison with our method.

### N.3. Recurrent Marginal Structural Networks (RMSNs)

Recurrent Marginal Structural Networks (RMSNs) (Lim, 2018) is a deep learning-based method to forecast counterfactual outcomes. Different from the MSMs, RMSNs employ the RNN model to build sequence-to-sequence architectures for counterfactual outcome prediction. To be specific, (Lim, 2018) employs RNNs to construct propensity networks and prediction networks, and combines these modules to form the inverse probability of treatment weighting (IPTW) estimation. In our configurations, we utilize RMSNs to fit outcomes on treatments and covariates.

### N.4. Causal Forest

Causal Forest (Wager & Athey, 2018) is a random forest-based method for heterogeneous treatment effects estimation, the treatment effects are estimated at the leaves of the random trees. We employ a Python library called EconML https://econml.azurewebsites.net/ to realize the Causal Forest. Although EconML does not support the estimation of the counterfactual outcomes directly, for most estimators in EconML, we can combine a baseline predictive model with one estimator in EconML to construct the counterfactual outcomes estimation. Specifically, we use a regression model to fit treatments on outcomes, and then train the causal forest model to estimate the treatment effects. Finally, we combine the causal forest and regression model to build the counterfactual outcomes.

### N.5. Linear Regression (LR)

We develop a linear regression-based method as an additional baseline. Since LR cannot directly handle matrix data such as spatial point patterns, we transform all treatments ($z_t$) and outcomes ($Y_t$) into the number of events contained in the treatments and outcomes ($R(z_t)$ and $R(Y_t)$), $R()$ is the dimension reduction map defined in Eq. (5) in the main text. Therefore, the data used for LR are all scalar. Subsequently, we use $R(z_1)$, $R(z_2)$,..., $R(z_M)$ to regress $R(Y_M)$, and the regression model obtained is denoted as $E[R(Y_M)|R(z_{\leq M})]$. After obtaining the regression model, we replace the independent variables, $R(z_1)$, $R(z_2)$,..., $R(z_M)$ in the regression model $E[R(Y_M)|R(z_{\leq M})]$ with $c * log(R(z_1))$, $c * log(R(z_2))$,..., $c * log(R(z_M))$, and then input them into the model to obtain the predicted values $\hat{N}_M^{\omega}(F_H)$. Similarly, we can obtain estimates at time $M + 1$. We use $R(z_1)$, $R(z_2)$,..., $R(z_{M+1})$ to regress $R(Y_{M+1})$, and the regression model obtained is denoted as $E[R(Y_{M+1})|R(z_{\leq M+1})]$. After obtaining the regression model, we replace the independent variables, $R(z_1)$, $R(z_2)$,..., $R(z_{M+1})$ in the regression model $E[R(Y_{M+1})|R(z_{\leq M+1})]$ with $c * log(R(z_1))$, $c * log(R(z_2))$,..., $c * log(R(z_{M+1}))$, and then input them into the model to obtain the predicted values $\hat{N}_{M+1}^{\omega}(F_H)$. In this way, we repeat the process until we obtain the estimate at the final time T, $\hat{N}_T^{\omega}(F_H)$. Then we can obtain estimates in the main text: $\hat{N}_{\omega}(F_H) = \frac{1}{T-M+1} \sum_{t=M}^{T} \hat{N}_t^{\omega}(F_H)$. We use the torch.nn module in PyTorch to implement linear regression, with 10 epochs and a learning rate set to 0.00004.

## O. Limitations

In this section, we discuss the limitations of our work. We employ the unconfoundedness assumption to identify our estimands and prove the propositions. The unconfoundedness assumption excludes the influences of unmeasured covariates. In real-world scenarios, unmeasured covariates may exist thus this assumption may be violated. However, the unmeasured covariates are still an open problem (Pearl, 2010), and the unconfoundedness assumption is still widely used in causal inference. We should be open to the unconfoundedness assumption.

