# OpenReview forum: "Transformer-Based Spatial-Temporal Counterfactual Outcomes Estimation"
_ICML.cc/2025/Conference — ICML 2025 poster_

### Official Review · Reviewer_Uw2a · 2025-03-12

**Overall Recommendation:** 3

**Summary:**

The paper studied counterfactual outcomes with spatial-tempora attributes using transformers and proved consistency and aymptotic normality of the estimator. The authors also conducted synthetic experiments and studied forest loss in Colombia. This paper is generally well-written.

*Edit after rebuttal: I changed score from 2 to 3.*

**Claims And Evidence:**

The authors claimed that using transformers to estimate the intensity function under the spatial Poisson assumption outperformed baseline estimators.

**Essential References Not Discussed:**

Satisfactory.

**Experimental Designs Or Analyses:**

The authors are very clear on the baseline and evaluation metrics. However, I think the lack of validation for the non-synthetic studies makes the story less convincing. I understand that there is non "true answer" for the non-synthetic studies, but the authors could perhaps perform sensitivity analysis or convergence analysis of their estimators.

I think it is helpful that the authors discuss the computation cost but a comparison of computation cost with the baseline might shed more light on the comparison between the proposed method.

**Methods And Evaluation Criteria:**

The authors tested their method on a synthetic Poission point process as well as Colombia forest change data and UCDP georeferenced event dataset. The authors used relative error rate for the synthetic experiment and checked the consistency of their conlusions with prior work as well as human judgement for the non-synthetic sutdies.

**Other Comments Or Suggestions:**

In the figures/tables of empirical studies, perhaps adds error bars and std/confidence bands.

**Other Strengths And Weaknesses:**

I think, despite this paper is generally well-written, the constribution seems to be a bit weak -- there is no significnat contributions on the theory front, and this paper has a more application flavor. However, the empirical studies and analyses are related limited compared with what I would expect from an application paper.

**Questions For Authors:**

Did you try to relax the Poisson assumption and see how well this method holds up in the synthetic setup?

**Relation To Broader Scientific Literature:**

Satisfactory.

**Theoretical Claims:**

This paper recalled known results on propensity scores, and proved a consistency result for the spacial estimator $N _ {\omega} (Y_t)$. While I think recalling the known results is helpful for the structure, I failed to be convinced that there is significnat novelty on the theoretical front -- the proof in Appendix B2 is thorough, but Proposition 3 seems to be more like a routine proof. To be fair, the authors did not claim theoretical contributions, so this is fine by me.

---

> ### Author Rebuttal · Authors · 2025-04-01
>
> ## Response to reviewer Uw2a
>
> We are glad the reviewer found our paper well-written. We would like to respond to each detailed point individually.
>
> >1. I failed to be convinced that there is significnat novelty on the theoretical front -- the proof in Appendix B2 is thorough, but Proposition 3 seems to be more like a routine proof. To be fair, the authors did not claim theoretical contributions, so this is fine by me.
>
> We appreciate that the reviewer found our proofs thorough. Now, we elaborate on our key contributions.
> One of our **key contributions** is leveraging deep learning to address the challenge of counterfactual outcome estimation for spatial-temporal data. Previous baselines are **unable** to achieve this. Specifically, our proposed method enables the estimation of propensity scores for spatial-temporal data. This **provides a feasible and efficient technical pathway** for causal inference in high-dimensional spatial-temporal settings. Besides, our theoretical analysis provides a guarantee for the proposed deep learning method.
>
> ---
>
> >2. However, I think the lack of validation for the non-synthetic studies makes the story less convincing. I understand that there is non "true answer" for the non-synthetic studies, but the authors could perhaps perform sensitivity analysis or convergence analysis of their estimators.
>
> We appreciate the reviewer's suggestions on sensitivity analysis and convergence analysis. Following your advice, we provide the theoretical sensitivity analysis in the **"sensitivity.pdf"** file in this anonymous [URL](https://anonymous.4open.science/r/DeppSTCI_Release_Version-master-3C91).
>
> With regard to the convergence analysis, Proposition 3 demonstrates that as time progresses, our estimator converges to the estimands (ground truth).
>
> ---
>
> >3. I think it is helpful that the authors discuss the computation cost but a comparison of computation cost with the baseline might shed more light on the comparison between the proposed method.
>
> We appreciate the suggestion to compare the computational cost with the baselines. All baselines were also run on an NVIDIA RTX 4090 GPU with an Intel Core i7-14700KF processor. For detailed information, please refer to the table below.
>
> Note that these baselines were **not designed for spatial-temporal data** and cannot process such data. Therefore, we convert all spatial-temporal data into scalar values for processing by these baselines (Appendix K.1). As a result, although our method does not significantly outperform these baselines in computation time, it **addresses a problem that they cannot solve.**
>
> | Methods | Time required for one setting |
> | --- | --- |
> | Ours | about 10 minutes |
> | LR | about 1.13 minutes |
> | MSMs | about  1.45 minutes |
> | RMSNs | about 1.53 minutes |
> |Causal Forest| about 2.25 minutes |
>
> ---
>
> >4. I think, despite this paper is generally well-written, the constribution seems to be a bit weak -- there is no significnat contributions on the theory front. However, the empirical studies and analyses are related limited compared with what I would expect from an application paper.
>
> We appreciate that the reviewer found our work well-written. Our main contribution lies in **addressing the challenge** of counterfactual outcome estimation for spatial-temporal data, which previous baselines are **unable to solve.** We provide a **feasible and efficient technical pathway** for causal inference in high-dimensional spatial-temporal data. The theoretical section provides the **foundation** for the proposed method.
>
> ---
>
> >5. In the figures/tables of empirical studies, perhaps adds error bars and std/confidence bands.
>
> We appreciate the suggestion to include error bars and confidence intervals. However, in the additional experiments, we trained the neural networks 25 times with different training data, and the results were consistently superior. Therefore, we believe the additional experiments demonstrate the robustness of our method.
>
> ---
>
> >6. Did you try to relax the Poisson assumption and see how well this method holds up in the synthetic setup?
>
> We acknowledge that the Poisson assumption may be somewhat restrictive. In this work, we use the spatial Poisson point process as a fundamental hypothesis for modeling spatial-temporal data. Therefore, if the Poisson assumption is violated, our theoretical framework would need to be restructured, and its performance may be compromised. In statistics, modeling spatial data distributions without assuming a specific distribution remains an open problem, and the Poisson assumption is widely used to describe spatial events [1-2].
>
> [1] Cressie, N., & Wikle, C. K. (2011). Statistics for spatio-temporal data. John Wiley & Sons.
>
> [2] Cressie, N. (2015). Statistics for spatial data. John Wiley & Sons.

---

> > ### Comment · Reviewer_Uw2a · 2025-04-09
> >
> > Thank you authors for addressing my questions and provide more details on computational costs and the write-up on sensitivity analysis. I still think this manuscript needs more work but won't hold back if other reviewers would like to champion it.

---

> > > ### Author Response · Authors · 2025-04-09
> > >
> > > Dear Reviewer Uw2a,
> > >
> > > Thanks a lot for your comments, and for raising the score.
> > >
> > > Now we articulate the key contributions of our work. In this paper, we propose a feasible and efficient technical pathway for causal inference in high-dimensional spatial-temporal data. In addition, we provide the theoretical guarantee for the proposed method.
> > >
> > > Following your advice, we will **take the following measures to enhance our manuscript.**
> > > * First, we will repeat all our experiments twenty times and add the error bars to the final version of our paper.
> > > * Second, we will relax the Poisson assumption and examine the performance of our method. All results will be added to the final version of our paper.
> > >
> > > Thank you once again for your invaluable contribution to the enhancement of our research.
> > >
> > > Best regards,
> > >
> > > Authors

---

### Official Review · Reviewer_pyND · 2025-03-15

**Overall Recommendation:** 3

**Summary:**

This paper introduces a Transformer-based framework for counterfactual outcome estimation in spatial-temporal data. It aims to improve causal inference in settings where treatments and outcomes are structured across both space and time. The authors propose a novel deep-learning-based estimator with a CNN-Based Propensity Score Estimation and Transformer-Based Intensity Function Estimation. The framework was supported by both theoretical insights and experiments on synthetic and real data.

**Claims And Evidence:**

The theoretical evidence is adundant, whereas the empirical results lack sufficient ablation to support the claims such as the improvements from the CNN based propensity score estimation.

**Essential References Not Discussed:**

I would encourage including more references to the temporal counterfactual prediction literature as it is an active research area, including (but not limited to):

Seedat, N., Imrie, F., Bellot, A., Qian, Z., & van der Schaar, M. (2022). Continuous-time modeling of counterfactual outcomes using neural controlled differential equations. arXiv preprint arXiv:2206.08311.

Wu, S., Zhou, W., Chen, M., & Zhu, S. (2024). Counterfactual generative models for time-varying treatments. In Proceedings of the 30th ACM SIGKDD Conference on Knowledge Discovery and Data Mining (pp. 3402-3413).

El Bouchattaoui, M., Tami, M., Lepetit, B., & Cournède, P. H. (2024). Causal contrastive learning for counterfactual regression over time. Advances in Neural Information Processing Systems, 37, 1333-1369.

**Experimental Designs Or Analyses:**

See below.

**Methods And Evaluation Criteria:**

Methods do make sense, but the evaluation criteria is lacking.

**Other Comments Or Suggestions:**

No

**Other Strengths And Weaknesses:**

Strengths
One thing that stands out is the consistent and rigorous theoretical framework of the work, which can potentially lay the foundation for future works on spatio-temporal causal inference.

Weaknesses
The experiment section is lacking, especially:
1. The details of the transformer architecture seems to be missing: what positional encoding are you using? what tokenization scheme is used?
2. More ablation is needed to identify the contribution of each innovation, including the CNN-based propensity score estimator.
3. Following my last point, it seems that when comparing to other baselines, it is unclear if the performance gain come from model backbone (i.e. transformer) or the proposed spatio-temporal causal inference framework. One way is to keep the model backbone simple for similar to other baselines, and adopt several ablation studies to demonstrate the gains brought by the two major innovations.

**Questions For Authors:**

See above.

**Relation To Broader Scientific Literature:**

Spatio-temporal causal inference is very important and broadly applicable.

**Theoretical Claims:**

I didn't check all the details. The proposals and theorems look good.

---

> ### Author Rebuttal · Authors · 2025-04-01
>
> ## Response to reviewer pyND
>
> We are glad the reviewer found the theoretical framework rigorous. We would like to respond to each detailed point individually.
>
> >1. The theoretical evidence is adundant, whereas the empirical results lack sufficient ablation to support the claims such as the improvements from the CNN based propensity score estimation.
>
> We really appreciate that the reviewer found the theoretical evidence abundant. Now we explain why removing the propensity score estimation is inappropriate in our setting.
>
> In our work, the CNN-based propensity score estimation serves as a practical implementation of the propensity score in our estimator framework (Sec. 4.4). Therefore, if we remove it, we effectively eliminate the propensity score from the estimator framework. As a result, the estimator would be **theoretically incomplete** and unable to perform its intended function properly.
>
> ---
>
> >2. Methods do make sense, but the evaluation criteria is lacking.
>
> We are glad the reviewer found the methods reasonable. Now we justify our choice of evaluation criteria. In the simulation experiments, we use the relative error rate as a direct measure of the estimator’s accuracy. In real-world data experiments, since the ground truth is unavailable, we assess our approach by comparing our findings with those of existing studies.
>
> ---
>
> >3. Spatio-temporal causal inference is very important and broadly applicable.
>
> We are glad the reviewer found the studied problem important.
>
> ---
>
> >4. I would encourage including more references to the temporal counterfactual prediction literature as it is an active research area, including (but not limited to).
>
> We really appreciate the reviewer's suggestions on the references. Following your advice, we **have cited and discussed** these excellent works in our paper.
>
> **Temporal Counterfactual Prediction**
> Temporal counterfactual prediction refers to performing counterfactual outcome prediction under time-varying settings. Seedat, N. et al. [1] propose TE-CDE, a neural-controlled differential equation approach for counterfactual outcome estimation in irregularly sampled time-series data. Wu, S. et al. [2] introduce a conditional generative framework for counterfactual outcome estimation under time-varying treatments, addressing challenges in high-dimensional outcomes and distribution mismatch. El Bouchattaoui, M. et al. [3] propose an RNN-based approach for counterfactual regression over time, focusing on long-term treatment effect estimation.
>
> [1] Seedat, N. et al. Continuous-time modeling of counterfactual outcomes using neural controlled differential equations.
>
> [2] Wu, S. et al. Counterfactual generative models for time-varying treatments.
>
> [3] El Bouchattaoui, M. et al. Causal contrastive learning for counterfactual regression over time.
>
> ---
>
> >5. The details of the transformer architecture seems to be missing: what positional encoding are you using? what tokenization scheme is used?
>
> We employ the Transformer to capture spatial information within a single time step. As a result, we do not apply fixed positional encoding.
>
> For the tokenization scheme, the Transformer processes discrete data points as input. Therefore, we treat each discrete coordinate as a token and map it to a high-dimensional embedding space.
>
> ---
>
> >6. More ablation is needed to identify the contribution of each innovation, including the CNN-based propensity score estimator.
>
> We appreciate the suggestion of the ablation of the CNN-based estimator. However, the ablation of the CNN-based estimator is inappropriate in our setting, for detailed reasons, please refer to response 1.
>
> ---
>
> >7. It is unclear if the performance gain come from model backbone (i.e. transformer) or the proposed spatio-temporal causal inference framework. One way is to keep the model backbone simple for similar to other baselines, and adopt several ablation studies to demonstrate the gains brought by the two major innovations.
>
> Compared to previous baselines, our work primarily tackles a problem that these methods cannot handle. This is because the frameworks and implementations of these baselines are not applicable to spatial-temporal data. Therefore, we attribute the superior performance of our method to both its theoretical framework and model implementation.
>
> We appreciate the reviewer's suggestions on the ablation studies. However, it is not appropriate to maintain the same model backbone as other baselines. For example, we employ a CNN to process the high-dimensional tensor. If we replace the CNN with the simple logistic regression or linear regression used in MSMs, the propensity score in our work would not be estimable.

---

> > ### Comment · Reviewer_pyND · 2025-04-02
> >
> > I've read the comments and most of my concerns are addressed. Raising the score from 2 to 3. I would still strongly recommend supplementing more experimental results and add the aforementioned clarification into the main text/supplements.

---

> > > ### Author Response · Authors · 2025-04-03
> > >
> > > Dear Reviewer pyND,
> > >
> > > Thanks a lot for your comments, and for raising the score.
> > >
> > > We sincerely appreciate your support for our paper and are particularly grateful for your invaluable comments.
> > >
> > > Following your advice, we will add experiments **replacing the Transformer with RNNs** to better demonstrate the gains brought by the major innovations. We will also include the results in the final version of the paper. Additionally, we commit to incorporating the aforementioned clarification into the main text or the appendix.
> > >
> > > Thank you once again for your invaluable contribution to the enhancement of our research.
> > >
> > > Best regards,
> > >
> > > Authors

---

### Official Review · Reviewer_p6Zj · 2025-03-17

**Overall Recommendation:** 4

**Summary:**

The paper introduces an approach to estimate counterfactual outcomes in a spatial-temporal setting, where both treatment and outcome may be represented in a high-dimensional space. The proposed method adapts IPW to the spatial-temporal setting, leveraging propensity score. The approach is implemented in practice using a deep learning architecture, leveraging CNNs for computing propensity scores and transformers for encoding the intensity function. Finally, the method is compared empirically with traditional counterfactual outcomes estimation approaches and achieves significantly lower error in high-dimensional settings.

**Claims And Evidence:**

The paper makes three claims: (1) they propose an estimator for counterfactual outcomes in the spatial-temporal setting; (2) they use a deep-learning approach to perform this estimation, leveraging CNNs for propensity score calculation and transformers for modeling intensity functions; and (3) the approach is demonstrated to outperform baselines empirically. See below sections for the evaluation of these claims.

**Essential References Not Discussed:**

N/A

**Experimental Designs Or Analyses:**

The proposed architecture is tested in both synthetic and real data, compared to baselines which do not take the spatial-temporal nature of the data into account. The results seem to convincingly show that the proposed architecture works better in these data settings.

**Methods And Evaluation Criteria:**

The data is separated into treatment $Z$, outcome $Y$, and covariates $X$. Importantly, each variable is collected across several time steps, and $Z$ and $Y$ are modeled as spatial point processes rather than traditional binary treatments and outcomes. An inverse probability weighting estimator is derived, composed of the propensity score, the counterfactual probability, and the intensity function of the outcome. The estimator estimates the expected number of outcomes in a specific region at a specific time. This estimator (Eq. 3) seems to be the main contribution of the paper, but it is unclear why this is the quantity of interest and how the estimator is derived.

The estimator is implemented in practice using a CNN to compute the propensity score and a transformer to compute the intensity function. The architecture seems to make sense for the task.

**Other Comments Or Suggestions:**

N/A

**Other Strengths And Weaknesses:**

I appreciate that the assumptions are clearly highlighted.

I think that the clarity of the paper is its greatest weakness. The paper uses a lot of convoluted notation and does not explain much of the math. It would be much clearer if the paper walked through a running example, where each term of the estimand is clearly highlighted.

**Questions For Authors:**

Why is the dimensionality reduction in Eq. 5 possible? Are we not losing information?

**Relation To Broader Scientific Literature:**

The related works section adequately sets the background of the task.

**Theoretical Claims:**

The paper proves three propositions in Sec. 4.5 about the estimator in Eq. 3. The proofs seem to hold, but it seems generally unclear how to interpret these results and why they are important. Prop. 3 seems to be the most important as it seems to indicate that the estimator properly converges to the intended value, but there seems to be some subtleties in the result that could be elaborated.

---

> ### Author Rebuttal · Authors · 2025-04-01
>
> ## Response to reviewer p6Zj
>
> We greatly appreciate the reviewer's comments and suggestions. We would like to respond to each detailed point individually.
>
> > 1. The estimator estimates the expected number of outcomes in a specific region at a specific time. This estimator (Eq. 3) seems to be the main contribution of the paper, but it is unclear why this is the quantity of interest and how the estimator is derived.
>
> Now we explain why the expected number of outcomes in a specific region at a specific time is the quantity of interest. These estimands provide spatial-temporal decision-making evidence for policy makers. For example, during epidemic control, policy makers can evaluate the average number of infections in specific areas at specific times under different isolation strategies, thereby optimizing the allocation of isolation resources.
>
> Next, we briefly introduce how the estimator is derived. The estimator is derived based on the Inverse Probability Weighting (IPW) strategy in Marginal Structural Models (MSMs) [1]. Specifically, the $\lambda_{Y_t^{ob}(z_{\leq t})}(s)$ in Eq.3 is the intensity function of outcomes in time $t$ and location $s$, and $\prod_{j=t-M+1}^{t} \frac{p_{h_j}(z_j)}{e_j(z_j)}$ represents the weights similar to those in the MSMs. In summary, we adopt the weighting strategy of MSMs to the studied spatial-temporal setting.
>
> [1] Robins J M, Hernan M A, Brumback B. Marginal structural models and causal inference in epidemiology[J]. Epidemiology, 2000, 11(5): 550-560.
>
> ---
>
> >2. The proofs seem to hold, but it seems generally unclear how to interpret these results and why they are important. Prop. 3 seems to be the most important as it seems to indicate that the estimator properly converges to the intended value, but there seems to be some subtleties in the result that could be elaborated.
>
>
> Prop. 1 and Prop. 2 are two essential properties of the propensity score. Specifically, these propositions demonstrate that the propensity score can make the estimation more **accurate and efficient**. Therefore, by proving Prop. 1 and Prop. 2, we validate that the propensity score used for spatial-temporal data is "reasonable" and can assist in counterfactual outcome estimation.
>
>
> Prop. 3 establishes the **theoretical reliability** of the estimator. It demonstrates that the expected value of the proposed estimator equals the estimands (i.e., unbiasedness) and that the variance of the estimation error converges to a finite value.
>
> ---
>
> >3. I appreciate that the assumptions are clearly highlighted.
>
> We are glad the highlighted assumptions are helpful.
>
> ---
>
> >4. I think that the clarity of the paper is its greatest weakness. It would be much clearer if the paper walked through a running example, where each term of the estimand is clearly highlighted.
>
> We appreciate the reviewer's suggestion on the running example. Following your advice, we provide an example that illustrates the estimands.
>
> For simplicity, consider the case of $t=8$ and $M=3$. Then the estimands
> $$
> N^\omega_8(F_H) =  \int_{Z^3} |S_{Y_8^{ob}(z_{\leq 8}(F_H))} \cap \omega| dF_H(z_{\left[ \text{6,8} \right]}),
> $$
> represent the expected number of outcomes in $t=8$ and region $\omega$ under distribution $F_H$. Next, we employ a table to elaborate on each term of the estimands.
>
>
> | Term | Interpretation |
> | --- | --- |
> | $t$=8, $M$=3 | Evaluates outcomes at time 8, considering 3-times intervention persistence (times 6-8). |
> | $z_{[6,8]}$ | Sequence of treatment variables over the time window [6,8]. |
> |$F_H(z_{\left[ \text{6,8} \right]})$  | Joint probability distribution of $z_{[6,8]}$ under counterfactual intervention strategy $F_H$.|
> | &vert;$S_{Y_8^{ob}(z_{\leq 8}(F_H))} \cap \omega$&vert; | Observed outcome counts in region $\omega$ at time 8, under $F_H$.  |
> | $Z^3$ | All possible values of $z_{[6,8]}$.  |
> | $\int_{Z^3}\cdot dF_H(z_{\left[ \text{6,8} \right]})$  | Expectation computation over all possible values of $z_{[6,8]}$. |
>
> ---
>
> >5. Why is the dimensionality reduction in Eq. 5 possible? Are we not losing information?
>
> In this work, we mainly estimate the outcome quantities. Therefore, we believe it is possible to reduce the studied variables to their quantity. We acknowledge that this strategy may result in some information loss; however, it is **useful for estimating the number of outcome events**. Moreover, this strategy **has its own theoretical significance** in terms of how to reduce the dimensionality for complicated problems.

---

> > ### Comment · Reviewer_p6Zj · 2025-04-04
> >
> > Thank you for addressing my points in your rebuttal. A lot of the results have been made more clear for me. Can you elaborate on " this strategy has its own theoretical significance in terms of how to reduce the dimensionality for complicated problems"?

---

> > > ### Author Response · Authors · 2025-04-05
> > >
> > > Dear Reviewer p6Zj,
> > >
> > > Thanks a lot for your acknowledgement and reply.
> > >
> > > Now we briefly explain "this strategy has its own theoretical significance ...". Eq. 5 maps the treatment variable set in time $t$ and region $\Omega$ ($\lbrace Z_t(s) | s \in \Omega \rbrace$) to the count of treated locations (scalar). This strategy retains sufficient information for estimating outcome quantities (total events) while avoiding intractable computations.
> > >
> > > This strategy is grounded in our problem. Since we assume treatments are generated by the spatial Poisson process (Assumption 2), the count of treated locations naturally follows a Poisson distribution. Therefore, the distribution of the propensity scores is specified and can be calculated.
> > >
> > > We hope this clarifies the rationale behind Eq. 5 and its role in our framework.
> > >
> > > Best regards,
> > >
> > > Authors

---

### Decision · Program_Chairs · 2025-05-01

**Decision:**

Accept (poster)

**Comment:**

The paper addresses the challenge of estimating counterfactual outcomes from observational data in a spatial-temporal context, where treatment and outcome variables are both influenced by spatial and temporal factors. The proposed method operates under the assumption that there is no unmeasured confounding present in the observations. It utilizes propensity scores to adapt the well-known Inverse Probability Weighting (IPW) method to this spatial-temporal setting.

To implement this approach, the authors employ a deep learning architecture that combines Convolutional Neural Networks (CNNs) for computing propensity scores with transformers for encoding the intensity function. The effectiveness of this method is demonstrated through empirical comparisons with traditional counterfactual outcome estimation approaches, showing a reduction in error.

Overall, this paper expands on a classic IPW algorithm, applying it to high-dimensional, spatial-temporal data. It is particularly relevant for researchers and practitioners in reinforcement learning and causal inference. However, since this paper explores a novel domain, readers may find it challenging to visualize practical applications for causal effect estimation in spatial-temporal contexts. Therefore, it would be encouraged for the authors to include a running example that illustrates the proposed method throughout the paper.